# Gamma: Toward Generic Image Assessment with Mixture of Assessment Experts

## Abstract

Image assessment aims to evaluate the quality and aesthetics of images and has been applied across various scenarios, such as natural and AIGC scenes. Existing methods mostly address these sub-tasks or scenes individually. While some works attempt to develop unified image assessment models, they have struggled to achieve satisfactory performance or cover a broad spectrum of assessment scenarios. In this paper, we present **Gamma**, a **G**eneric im**A**ge assess**M**ent model using **M**ixture of **A**ssessment Experts, which can effectively assess images from diverse scenes through mixed-dataset training. Achieving unified training in image assessment presents significant challenges due to annotation biases across different datasets. To address this issue, we first propose a Mixture of Assessment Experts (MoAE) module, which employs shared and adaptive experts to dynamically learn common and specific knowledge for different datasets, respectively. In addition, we introduce a Scene-based Differential Prompt (SDP) strategy, which uses scene-specific prompts to provide prior knowledge and guidance during the learning process, further boosting adaptation for various scenes. Our Gamma model is trained and evaluated on 12 datasets spanning 6 image assessment scenarios. Extensive experiments show that our unified Gamma outperforms other state-of-the-art mixed-training methods by significant margins while covering more scenes.

## 1 Introduction

Image assessment is a long-standing research topic in the field of image processing, primarily comprising two tasks: Image Quality Assessment (IQA) and Image Aesthetic Assessment (IAA). These tasks require models to automatically evaluate the visual quality and aesthetic appeal of images, respectively. They have broad applications in various real-world scenarios, such as guiding image dehazing (Zhao et al., 2021), selecting high-quality images in a data engine (Rombach et al., 2022), serving as tools in an agentic system (Yang et al., 2024), or acting as reward models when aligning image generative models with human feedback (Liang et al., 2024).

Due to differences in image content and application scenarios, image assessment has spawned many sub-tasks, such as Natural-IQA for natural images, Face-IQA for facial images, AIGC-IQA for generated images, and IAA. Accordingly, numerous methods (Ke et al., 2021; He et al., 2022; Su et al., 2023b) have been proposed to address these specific tasks. However, these models often struggle to apply directly to other scenes or typically require task-specific fine-tuning on a given dataset. As illustrated in Figure 1, it is challenging for DEIQT (Qin et al., 2023) to transfer to other datasets without fine-tuning. This limitation prevents image assessment models from being widely applicable, *e.g.*, IQA models are needed to assess facial, artistic, and natural images in the AIGC scene. Hence, there is an urgent need for a model that can effectively handle a variety of scenarios.

To mitigate this issue, some approaches attempt to combine many datasets from various assessment tasks to train a general image assessment model. For instance, UNIQUE (Zhang et al., 2021) and LIQE (Zhang et al., 2023) utilize multiple authentic and synthetic natural IQA datasets for mixed training, but they focus only on Natural-IQA. Q-Align (Wu et al., 2023) uses a large language model with billions of parameters to unify IQA and IAA tasks, but it has a low inference speed and focuses solely on natural images. Additionally, PromptIQA (Chen et al., 2024b) employs image-score pairs as prompts for quality predictions, but it is inflexible as it requires multiple images as references during inference. These methods, however, fail to achieve competitive performance compared to

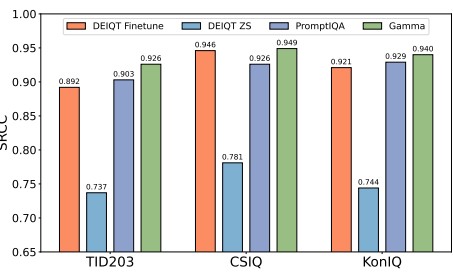

Figure 1: "DEIQT Finetune" is trained and tested on the same dataset. "DEIQT ZS" directly assesses images using the model trained on other datasets, which perform poorly. PromptIQA and our Gamma are trained on mixed datasets. Our Gamma performs well on multiple datasets simultaneously and even surpass the task-specific method DETQT.

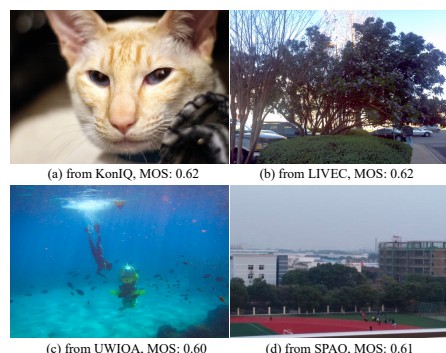

Figure 2: Images with similar MOS labels from different datasets exhibit drastically different perceptual quality. It is not hard to observe that image (a) has clearly superior quality than the other three. Zoom in for a better view.

models trained on specific datasets and cover a broad range of scenes. Thus, it is imperative to develop a foundational image assessment model capable of evaluating images from various scenes.

To this end, we present **Gamma**, a **G**eneric im**A**ge assess**M**ent model using **M**ixture of **A**ssessment experts, to achieve unified image assessment across multiple datasets effectively. We found that the primary challenge in mixed-dataset training is *the mean opinion score (MOS) bias* between different datasets, *e.g.*, images with similar MOS may exhibit different perceptual qualities across various datasets. As shown in Figure 2, samples from KonIQ (Hosu et al., 2020), LIVEC (Ghadiyaram & Bovik, 2015), UWIQA (Yang et al., 2021a), and SPAQ (Fang et al., 2020) show obvious differences in perceptual quality despite being labeled with similar MOS. To address this challenge, we introduce a novel **Mixture of Assessment Experts (MoAE)** module in our Gamma model. MoAE consists of two types of experts: shared experts and adaptive experts. The shared experts are employed throughout to learn dataset-shared knowledge, while the adaptive experts are dynamically activated to varying degrees to learn dataset-specific knowledge. Additionally, an image-based router modulates the contributions of each adaptive expert. This strategy allows the model to capture common features while flexibly adjusting representative features for different datasets. Compared with general Mixture of Experts (MoE) (Shazeer et al., 2017) and Lora-based MoE (Liu et al., 2024), we equip the MoAE module only in the rear blocks instead of all blocks, making it more efficient. Secondly, we propose a **Scene-based Differential Prompt (SDP)**, which uses different prompts for different datasets according to their scenes. This strategy provides scene-specific knowledge for representation learning of different datasets, guiding the mixed-dataset training process.

Our Gamma model is uniformly trained on a mixture of 12 datasets from 6 image assessment scenarios spanning IQA and IAA tasks. We then evaluate it on 12 datasets, demonstrating that it not only outperforms state-of-the-art (SOTA) mixed-training methods by notable margins, but also covers more scenarios. In some benchmarks, Gamma even surpasses some SOTA task-specific methods. Additionally, if we fine-tune our MoAE-equipped Gamma on specific datasets, it can achieve SOTA performance on 12 datasets. Moreover, the unified pre-trained Gamma can be utilized as a foundational model to significantly enhance other image assessment tasks, *e.g.*, medical image quality assessment, and can achieve SOTA performance after task-specific training. Our contributions can be summarized as follows:

- We present **Gamma**, a powerful and generic image assessment model, capable of accurately assessing images from various scenarios through mixed training.

- We propose a novel Mixture of Assessment Experts (MoAE) module to extract representative features adaptively and a Scene-based Differential Prompt (SDP) strategy to provide guidance for representation learning, thereby achieving effective mixed-dataset training.

- Extensive experiments show that Gamma achieves SOTA performance on 12 datasets across 6 image assessment scenes in both mixed training and task-specific training settings.

## 2 RELATED WORK

### 2.1 IMAGE ASSESSMENT

Image Assessment mainly includes two subtasks: Image Quality Assessment (IQA) and Image Aesthetic Assessment (IAA). The IQA task focuses on the distortion level of the image, while IAA aims to evaluate the aesthetic appeal of the image. In the deep learning era, these two tasks have achieved significant breakthroughs. For the IQA task, researchers develop various advanced techniques to improve performance, including multi-level feature aggregation (Li et al., 2018; Chen et al., 2024a; Xu et al., 2024; Zhang et al., 2018; Mittal et al., 2012b; Ying et al., 2020), adaptive convolution (Su et al., 2020), transformer methods (Ke et al., 2021; Qin et al., 2023), vision-language models (VLMs) (Wang et al., 2023; Zhang et al., 2023) and large language models (LLM) (You et al., 2023). Moreover, besides the natural image assessment, these are various IQA methods for other scenes, such as face IQA (Ou et al., 2021; Su et al., 2023b; Jo et al., 2023), AIGC IQA (Yuan et al., 2023), underwater IQA (Yang et al., 2021b; Guo et al., 2023; Yang & Sowmya, 2015; Liu et al., 2023). For the IAA task, numerous methods have also been proposed to improve the model performance, including loss function (Talebi & Milanfar, 2018), novel transformer architecture (Tu et al., 2022), multi-level features (Hosu et al., 2019), theme information (He et al., 2022; Li et al., 2023b) and multimodal pre-training (Ke et al., 2023).

### 2.2 MIXED TRAINING FOR IMAGE ASSESSMENT

As a fundamental image processing task, image assessment has achieved remarkable success and has been applied to various scenarios. Recently, some works have attempted to develop unified methods that can be used in multiple IQA settings. To achieve this goal, one approach is to conduct mixed training across multiple IQA datasets. UNIQUE (Zhang et al., 2021) sampled pairs of images from IQA datasets and computes the probability that the first image of each pair is of higher quality. StairIQA (Sun et al., 2023) designed separate IQA regression heads for each dataset. PromptIQA (Chen et al., 2024b) utilized a short sequence of Image-Score Pairs as prompts for quality predictions. Q-Align (Wu et al., 2023) used large language model (LLM) to unify IQA and IAA tasks. However, most existing works fail to achieve competitive performance with task-specific methods and do not cover various scenes. This paper combines various datasets from both tasks and designs innovative modules to effectively learn a unified and generic image assessment perception.

### 2.3 MIXTURE OF EXPERTS

Mixture-of-Experts (MoE) divides specific parts of the parameters into several subsets, each of which is called an expert. It sets up a router that assigns experts to different inputs. Recently, the MoE structure has achieved remarkable success in large language models (LLM). For instance, DeepSeekMoE (Dai et al., 2024) proposes a novel MoE architecture that uses shared and routed experts to extract common and dynamic knowledge simultaneously. Beyond the natural language processing tasks, the idea of MoE has also been applied to vision models (Dai et al., 2021; Riquelme et al., 2021; Chen et al., 2023) and multimodal transformers (Wang et al., 2022; Feng et al., 2023). In Gamma, we utilize MoE to effectively learn specific and general features of multi-dataset.

## 3 METHOD

### 3.1 PRELIMINARY

As a foundational vision-language model (VLM), CLIP (Radford et al., 2021) has shown significant promise in supporting a wide array of vision tasks. Specifically, CLIP is composed of a transformer-based visual encoder $\mathcal{V}$ and a text encoder $\mathcal{T}$, which generate aligned visual representations $I$ and text representations $T$ for each image-text pair. Utilizing these features, we can compute cosine similarity scores between image and text pairs across different domains or tasks to perform task-specific predictions, including image assessment tasks. Recently, to enhance CLIP's capabilities in the field of image assessment, UniQA (Zhou et al., 2024) fine-tuned CLIP on large-scale synthetic and authentic image-text datasets focused on image quality and aesthetics. This approach demonstrates excellent performance on both IQA and IAA tasks after task-specific fine-tuning. However,

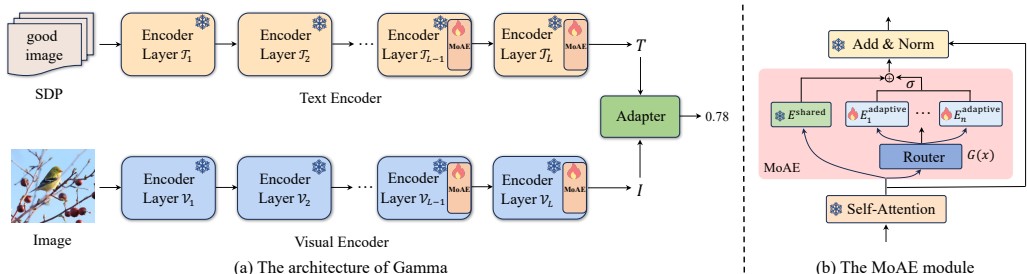

Figure 3: (a) The architecture of Gamma: It consists of a visual encoder $\mathcal{V}$ and text encoder $\mathcal{T}$; We add the Mixture of Assessment Experts (MoAE) to the last few layers of both encoders. We introduce a Scene-based Differential Prompt (SDP) to prompt images from different scenes (See Section 3.4 for details). (b) The MoAE module: It involves one shared expert ($E^{\mathrm{shared}}$) and several adaptive experts (from $E_1^{\mathrm{adaptive}}$ to $E_n^{\mathrm{adaptive}}$). We employ a router to adaptively activate the adaptive experts. We then use a learnable factor $\sigma$ to merge the features of two type of experts.

the model lacks foundational applicability across various image assessment scenarios without fine-tuning. Building on the unified training pipeline proposed in UniQA, we propose two approaches to confront MOS bias present in different datasets and develop a foundational image assessment model. In the following, we will provide a detailed exposition of its components.

## 3.2 OVERVIEW OF GAMMA

As illustrated in Figure 3, our Gamma employs a visual encoder $\mathcal{V}$ to extract visual features $I \in \mathbb{R}^d$, and a text encoder $\mathcal{T}$ to extract text features $T \in \mathbb{R}^d$. After these encoders, a tunable adapter is used to obtain a score q representing image quality or aesthetics, following the methods in Zhang et al. (2023) and Zhou et al. (2024). This process can be described as:

$$\mathrm{q} = \sum_{k=1}^{5} \mathrm{C}_k \mathrm{Softmax}(I'^{\top} T_k / \tau), \qquad I' = \mathrm{Adapter}(I), \tag{1}$$

where $\{T_k\}_{k=1}^5 \in \mathbb{R}^{5 \times d}$ represents text features of five assessment-dependent text prompts, *e.g.*, {bad image,poor image,fair image,good image,perfect image}. $\{\mathrm{C}_k\}_{k=1}^5 \in \mathbb{R}^5$ is a learnable vector initialized to $[0.2, 0.4, 0.6, 0.8, 1.0]$, and $\tau$ is a temperature parameter. In practice, the $\mathrm{Adapter}(\cdot)$ consists of two fully connected layers with a $\mathrm{ReLU}(\cdot)$ activation function in between. Based on this structure, to confront MOS bias in the mixed dataset and effectively perform unified pre-training, we propose a **Mixture of Assessment Experts (MoAE)** module to adaptively learn dataset-shared and dataset-specific knowledge from different datasets. We just integrate the MoAE module into the last few layers of both encoders, as shown in Figure 3 (a). Notably, we only fine-tune the parameters of the MoAE modules for various tasks, keeping the other parameters frozen, which is a significant advantage of our method. Additionally, we introduce a **Scene-based Differential Prompt (SDP)**. It uses different prompts for datasets from different scenes, thereby providing useful scene-based guidance for mixed-dataset training.

## 3.3 MIXTURE OF ASSESSMENT EXPERTS

To develop a unified and generic image assessment model, we aim to combine multiple image assessment datasets for joint training. Unfortunately, the mean opinion score (MOS) introduces significant biases across different datasets, which hinders joint training. To address this challenge, we propose the MoAE module, where several experts are employed to learn the diverse biases of different datasets. As shown in Figure 3 (b), the proposed MoAE module includes a shared assessment expert ($E^{\mathrm{shared}}$) to learn common knowledge of image assessment and several adaptive assessment experts (from $E_1^{\mathrm{adaptive}}$ to $E_n^{\mathrm{adaptive}}$) to dynamically learn dataset-specific knowledge.

**The Shared Assessment Expert.** The shared assessment expert $E^{\mathrm{shared}}$ inherits the image assessment capabilities of the original CLIP model by reusing its weights. This expert remains frozen during training to ensure that the learned world knowledge is retained. Thus, the model can capture

common representations across various contexts and maintain its original multi-modal capabilities. Given an input hidden state $x \in \mathbb{R}^d$, the output of the shared assessment expert is:

$$y^{\text{shared}} = E^{\text{shared}}(x), \tag{2}$$

where $E^{\text{shared}}(\cdot)$ is implemented as the original feed-forward network (FFN) of the CLIP model.

**The Adaptive Assessment Expert.** The adaptive assessment expert module contains two components: (1) $n$ experts $\{E_i^{\text{adaptive}}\}_{i=1}^n$ to capture diverse facets of multi-dataset information; and (2) a router $G(\cdot)$ to tailor the contribution of different experts based on the input feature. Given an input feature $x \in \mathbb{R}^d$, the output $y^{\text{adaptive}}$ can be computed as:

$$y^{\text{adaptive}} = \sum_{i=1}^n G(x)_i E_i^{\text{adaptive}}(x), \qquad G(x) = \text{Softmax}(Wx). \tag{3}$$

Here, the router $G(\cdot)$ is a linear transformation for the input feature $x$; $W \in \mathbb{R}^{n \times d}$ is the transformation matrix. To avoid unreasonable weights, we utilize a $\text{Softmax}$ operator to normalize the contribution weights. This ensures that the model can learn dataset-specific knowledge efficiently.

**MoAE Module.** Based on the above experts, the MoAE module merges the features of the two types of experts with a learnable factor $\sigma$, as shown on the right side of Figure 3. Thus, the output of the MoAE module can be expressed as:

$$y^{\text{MoAE}} = y^{\text{shared}} + \sigma \cdot y^{\text{adaptive}}. \tag{4}$$

The $\sigma$ factor is zero-initialized so that the visual and text encoders can generate aligned features at the beginning. In practice, we freeze the shared experts and set the adaptive experts to be tunable only. This approach maintains parameter efficiency during mixed training and preserves the multi-modal capabilities of the original model.

We incorporate the MoAE module into the last $K$ layers of both visual and text encoders, as shown in Figure 3. This strategy makes our method both effective and efficient. The visual feature extraction process can be formulated as follows:

$$\begin{aligned} I_i &= \boldsymbol{\mathcal{V}}_i(I_{i-1}), \quad i = 1, 2, \ldots, L - K \\ I_j &= \boldsymbol{\mathcal{V}}_j^{\text{MoAE}}(I_{j-1}), \quad j = L - K + 1, \ldots, L \end{aligned} \tag{5}$$

where $L$ denotes the number of layers of the visual encoder; $I_i$ represents the visual features of the $i$-th encoder layer; and $\boldsymbol{\mathcal{V}}^{\text{MoAE}}$ represents the MoAE-equipped visual encoder layer. The text branch operates similarly to the visual branch.

### 3.4 Scene-based Differential Prompt

To facilitate scene-guided learning, we introduce a ***Scene-based Differential Prompt (SDP)*** to help the model acquire scene-specific knowledge from different datasets. We utilize 12 datasets spanning 6 image assessment scenarios, including synthetic distortion nature IQA, authentic distortion nature IQA, face IQA, AIGC IQA, underwater IQA, and IAA, for mixed training (details are recorded in Appendix A.2). We categorize these datasets into five groups based on their scenes: natural quality, AI-generated quality, underwater quality, face quality, and natural aesthetics. Specifically, for the face quality assessment dataset, we use prompts such as *face bad-quality, face poor-quality, face fair-quality, face good-quality, face perfect-quality* appended to the word *image*. For more details on these prompts, please refer to Appendix A.4.

This strategy effectively differentiates the feature space of images from various scenes and enhances scene-specific knowledge, thereby mitigating the MOS bias across different datasets. Experimental results show that this method significantly improves the model's performance (Table 1).

## 4 Experiments

### 4.1 Datasets and Evaluation Criteria

**Datasets.** We utilize 12 datasets for unified training and testing, encompassing both image quality and aesthetic assessment tasks. For the IQA task, five different assessment scenarios are included:

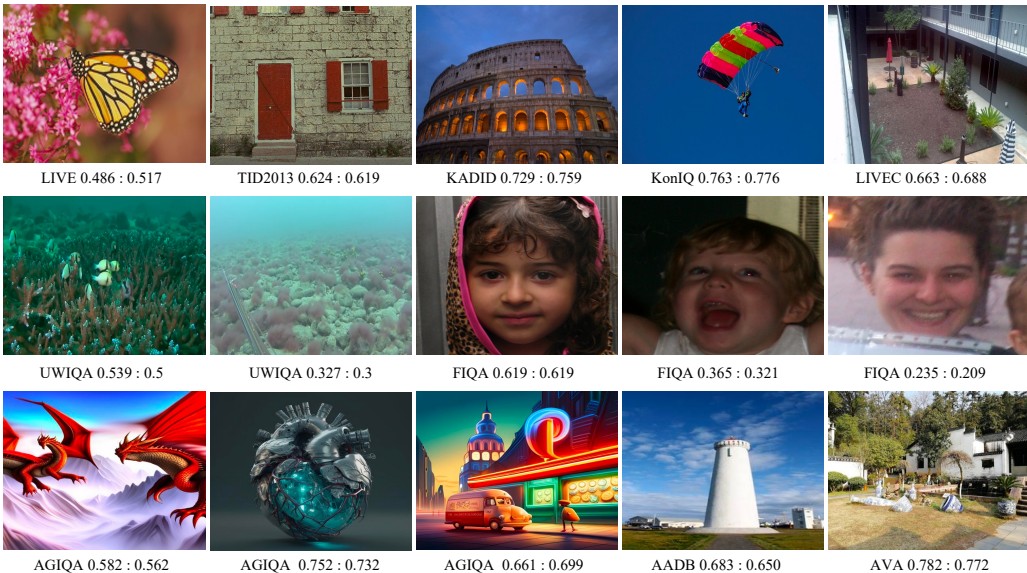

Figure 4: Visual examples from different datasets, which include natural images, underwater images, face images, AI-generated images, and *etc*. The first value is the prediction score and the second value is the ground-truth MOS. Our Gamma can accurately evaluate images from different scenes, demonstrating the generalization and effectiveness. All images are resized for better visibility.

synthetic distortion nature IQA (SDN-IQA), authentic distortion nature IQA (ADN-IQA), face IQA (F-IQA), AIGC IQA (AG-IQA), and underwater IQA (U-IQA). For the IAA task, we use two classical benchmarks, AVA and AADB. In addition, we use two rare datasets to verify the generalization ability of the model, *i.e.*, exBeDDE and ECIQAD. The exBeDDE is a dehazed image quality assessment (D-IQA) dataset, while ECIQAD is an enhanced colonoscopy image quality assessment (EC-IQA) dataset. Detailed information about these datasets is provided in Table 13.

**Evaluation Criteria.** We use Spearman's Rank-Order Correlation Coefficient (SRCC) and Pearson's Linear Correlation Coefficient (PLCC) as criteria to measure the performance of IQA models. Both coefficients range from 0 to 1, with higher values indicating better performance.

### 4.2 IMPLEMENTATION DETAILS

Following the settings in (Su et al., 2020; Ke et al., 2021), we randomly divide each dataset into 80% for training and 20% for testing. The training dataset is a mixture of the training sets of each dataset and we test Gamma on each test data separately. This process is repeated 10 times, and the median of the 10 scores is reported as the final score. We use the pre-trained weight of UniQA (Zhou et al., 2024), which uses CLIP-B/16 as multimodal encoder. We freeze the CLIP visual and text encoders, training only the MoAE module and adapter. For the unified training, we train the model for 10 epochs with a batch size of 8. The initial learning rate is set to 2e-5. We normalize the MOS/DMOS scale to [0, 1] for all datastes. We utilize Adam optimizer (Kingma, 2014) and MSE loss to optimize the model. For the task-specific training, we use different training settings according to the task and size of datasets. More training details are provided in the appendix.

### 4.3 MAIN RESULTS

Our MoAE-equipped model can be used for task-specific training and mixed training, both of which can achieve state-of-the-art (SOTA) performance, as shown in Table 1.

**Task-specific Training.** We apply our method to 12 image assessment datasets. We use the fixed naive prompt (described in Section 3.2) for training and testing. We observe that our method outperforms all others methods by a significant margin. On some benchmark, our method achieve dramatic improvements, such as SRCC of 0.944 (*v.s.* 0.916) on TID2013 and 0.945 (*v.s.* 0.933) on KonIQ.

Table 1: Comparison with SOTA task-specific and mixed-training models on 12 datasets for 6 image assessment tasks. "Gamma" and "Gamma-T" denote the mixed-training and task-specific models, respectively. Gamma$^\dagger$ uses the Scene-based Differential Prompt (SDP) for training and testing. $*$ indicates that we retrain the model with the same data split as ours.

| Task | | Synthetic Distortion Nature IQA (SDN-IQA) | | | | | | | | Authentic Distortion Nature IQA (ADN-IQA) | | | | | |
|---|---|---|---|---|---|---|---|---|---|---|---|---|---|---|---|
| Training | Dataset | LIVE | | CSIQ | | TID2013 | | KADID | | LIVEC | | KonIQ | | SPAQ | |
| Type | Method | SRCC | PLCC | SRCC | PLCC | SRCC | PLCC | SRCC | PLCC | SRCC | PLCC | SRCC | PLCC | SRCC | PLCC |
| Task Specific | HyperIQA | 0.962 | 0.966 | 0.923 | 0.942 | 0.840 | 0.858 | 0.852 | 0.845 | 0.859 | 0.882 | 0.906 | 0.917 | 0.911 | 0.915 |
| | MUSIQ | 0.940 | 0.911 | 0.871 | 0.893 | 0.773 | 0.815 | 0.875 | 0.872 | 0.702 | 0.746 | 0.916 | 0.928 | 0.918 | 0.921 |
| | TOPIQ | 0.943 | 0.942 | 0.908 | 0.925 | 0.813 | 0.845 | 0.877 | 0.875 | 0.833 | 0.868 | 0.915 | 0.925 | 0.914 | 0.917 |
| | DEIQT | 0.980 | 0.982 | 0.946 | 0.963 | 0.892 | 0.908 | 0.889 | 0.887 | 0.875 | 0.894 | 0.921 | 0.934 | 0.919 | 0.923 |
| | LoDA | 0.975 | 0.979 | - | - | 0.869 | 0.901 | 0.931 | 0.936 | 0.876 | 0.899 | 0.932 | 0.944 | 0.925 | 0.928 |
| | UniQA | 0.981 | **0.983** | 0.963 | 0.973 | 0.916 | 0.931 | 0.940 | 0.943 | 0.890 | 0.905 | 0.933 | 0.941 | 0.924 | 0.928 |
| | Gamma-T | **0.982** | 0.971 | **0.973** | **0.978** | **0.944** | **0.950** | 0.960 | 0.961 | 0.899 | 0.921 | 0.945 | 0.952 | 0.928 | 0.931 |
| Mixed Training | UNIQUE | 0.969 | **0.968** | 0.902 | 0.927 | - | - | 0.878 | 0.876 | 0.854 | 0.890 | 0.896 | 0.901 | - | - |
| | LIQE* | **0.972** | 0.953 | 0.946 | 0.943 | - | - | 0.932 | 0.933 | 0.902 | 0.908 | 0.920 | 0.905 | - | - |
| | StairIQA | 0.937 | 0.934 | 0.768 | 0.843 | 0.675 | 0.773 | 0.785 | 0.805 | 0.780 | 0.855 | 0.865 | 0.896 | 0.903 | 0.907 |
| | PromptIQA | 0.936 | 0.934 | 0.926 | 0.939 | 0.903 | 0.922 | 0.928 | 0.931 | **0.913** | **0.928** | 0.929 | 0.943 | 0.923 | 0.926 |
| | Gamma | 0.957 | 0.952 | 0.949 | 0.966 | 0.926 | 0.934 | 0.960 | 0.962 | 0.851 | 0.871 | **0.940** | 0.949 | 0.923 | 0.928 |
| | Gamma$^\dagger$ | 0.953 | 0.953 | **0.960** | **0.968** | **0.935** | **0.941** | **0.962** | **0.964** | 0.891 | 0.914 | 0.939 | **0.950** | 0.929 | 0.932 |

| Task | | Face IQA (F-IQA) | | AIGC IQA (AG-IQA) | | | Underwater IQA (U-IQA) | | | Image Aesthetic Assessment (IAA) | | | | | |
|---|---|---|---|---|---|---|---|---|---|---|---|---|---|---|---|
| Training | Dataset | GFIQA20k | | Dataset | AGIQA3k | | Dataset | UWIQA | | Dataset | AVA | | Dataset | AADB | |
| Type | Method | SRCC | PLCC | Method | SRCC | PLCC | Method | SRCC | PLCC | Method | SRCC | PLCC | Method | SRCC | PLCC |
| Task Specific | SDD-FIQA | 0.602 | 0.649 | DBCNN | 0.821 | 0.876 | FDUM | 0.694 | 0.689 | MaxViT | 0.708 | 0.745 | MUSIQ | 0.706 | 0.712 |
| | IFQA | 0.697 | 0.722 | HyperNet | 0.836 | 0.890 | UCIQE | 0.627 | 0.626 | TANet | 0.758 | 0.765 | TANet | 0.738 | 0.737 |
| | TOPIQ | 0.966 | 0.967 | CLIPIQA | 0.843 | 0.805 | URanker | 0.674 | 0.663 | VILA | 0.774 | 0.774 | TAVAR | 0.761 | 0.763 |
| | GPFIQA | 0.964 | 0.965 | PSCR | 0.850 | 0.906 | UIQI | 0.742 | 0.741 | UniQA | 0.776 | 0.776 | UniQA | 0.786 | 0.787 |
| | Gamma-T | **0.968** | **0.968** | Ours-T | **0.894** | **0.921** | Ours-T | **0.870** | **0.880** | Ours-T | **0.785** | **0.784** | Ours-T | **0.793** | **0.798** |
| Mixed Training | UNIQUE | - | - | UNIQUE | - | - | UNIQUE | - | - | UNIQUE | - | - | UNIQUE | - | - |
| | LIQE | - | - | LIQE | - | - | LIQE | - | - | LIQE | - | - | LIQE | - | - |
| | StairIQA | 0.937 | 0.935 | StairIQA | 0.755 | 0.833 | StairIQA | 0.722 | 0.727 | StairIQA | - | - | StairIQA | - | - |
| | PromptIQA | **0.970** | **0.971** | PromptIQA | 0.851 | 0.901 | PromptIQA | **0.877** | **0.884** | PromptIQA | - | - | PromptIQA | - | - |
| | Gamma | **0.970** | 0.970 | Gamma | 0.870 | 0.910 | Gamma | 0.863 | 0.878 | Gamma | 0.740 | 0.737 | Gamma | 0.742 | 0.743 |
| | Gamma$^\dagger$ | **0.970** | 0.970 | Gamma$^\dagger$ | **0.887** | **0.923** | Gamma$^\dagger$ | 0.873 | **0.884** | Gamma$^\dagger$ | **0.750** | **0.749** | Gamma$^\dagger$ | **0.756** | **0.755** |

Table 2: The effect of Mixture of Assessment Experts (MoAE) and Scene-based Differential Prompt (SDP). The MoAE and SDP can improve the performance of the model.

| Dataset | | LIVEC | | KonIQ | | LIVE | | CSIQ | | AGIQA3k | | UWIQA | | AVA | |
|---|---|---|---|---|---|---|---|---|---|---|---|---|---|---|---|
| MoAE | SDP | SRCC | PLCC | SRCC | PLCC | SRCC | PLCC | SRCC | PLCC | SRCC | PLCC | SRCC | PLCC | SRCC | PLCC |
| × | × | 0.765 | 0.792 | 0.858 | 0.885 | 0.927 | 0.918 | 0.852 | 0.898 | 0.800 | 0.866 | 0.750 | 0.768 | 0.681 | 0.672 |
| × | ✓ | 0.843 | 0.856 | 0.874 | 0.896 | 0.929 | 0.917 | 0.866 | 0.901 | 0.841 | 0.887 | 0.770 | 0.780 | 0.721 | 0.715 |
| ✓ | × | 0.851 | 0.871 | 0.940 | 0.949 | 0.957 | 0.952 | 0.949 | 0.966 | 0.870 | 0.910 | 0.863 | 0.878 | 0.740 | 0.737 |
| ✓ | ✓ | 0.891 | 0.914 | 0.939 | 0.950 | 0.960 | 0.968 | 0.953 | 0.953 | 0.887 | 0.923 | 0.873 | 0.884 | 0.750 | 0.749 |

Table 3: The impact of different number of experts in adaptive experts. We use naive prompt strategy for ablation.

| Dataset | LIVEC | | CSIQ | | TID2013 | | AGIQA3k | | UWIQA | |
|---|---|---|---|---|---|---|---|---|---|---|
| Experts Number | SRCC | PLCC | SRCC | PLCC | SRCC | PLCC | SRCC | PLCC | SRCC | PLCC |
| Zero Expert | 0.765 | 0.792 | 0.852 | 0.898 | 0.792 | 0.826 | 0.800 | 0.866 | 0.750 | 0.768 |
| One Expert | 0.842 | 0.866 | 0.945 | 0.963 | 0.918 | 0.931 | 0.866 | 0.908 | 0.859 | 0.873 |
| Three Experts | 0.851 | 0.871 | 0.949 | 0.966 | 0.926 | 0.934 | 0.870 | 0.910 | 0.863 | 0.878 |
| Five Experts | 0.854 | 0.889 | 0.951 | 0.965 | 0.926 | 0.934 | 0.872 | 0.911 | 0.860 | 0.876 |

Since images in these 12 datasets encompass a wide variety of contents and distortion types, it is particularly challenging to consistently achieve the leading performance on all of them.

**Mixed Training.** We conduct mixed training on 12 image assessment datasets. The trained model can be used to assess the images from these datasets. The experimental results are reported in Table 1. When compared with other mixed training models, such as StairIQA and PromptIQA, our method exhibits powerful and superior performance on each dataset. More importantly, our method can also be used to IAA tasks and demonstrates excellent performance. It is worth noting that our mixed training model even achieves results comparable to task-specific models on datasets such as KADID, KonIQA, SPAQ, and GFIQA. These results demonstrate that our approach can be effectively applied to different image assessment scenarios.

Table 4: The impact of different model configuration in the proposed MoAE module.

| Dataset | LIVEC | | CSIQ | | AGIQA3k | | UWIQA | | AVA | |
|---|---|---|---|---|---|---|---|---|---|---|
| Model Configuration | SRCC | PLCC | SRCC | PLCC | SRCC | PLCC | SRCC | PLCC | SRCC | PLCC |
| Unfreeze shared expert | 0.849 | 0.869 | 0.946 | 0.957 | 0.864 | 0.904 | 0.858 | 0.871 | 0.720 | 0.719 |
| w/o Merging factor $\sigma$ | 0.847 | 0.866 | 0.932 | 0.947 | 0.851 | 0.903 | 0.845 | 0.869 | 0.698 | 0.697 |
| Our MoAE module | 0.851 | 0.871 | 0.949 | 0.966 | 0.870 | 0.910 | 0.863 | 0.878 | 0.740 | 0.737 |

Table 5: The impact of adding MoAE to different numbers of layers for training.

| Dataset | Parms | FLOPs | LIVEC | | KonIQ | | LIVE | | CSIQ | | AGIQA3k | | UWIQA | | AVA | |
|---|---|---|---|---|---|---|---|---|---|---|---|---|---|---|---|---|
| MoE Layer | Million | Hours | SRCC | PLCC | SRCC | PLCC | SRCC | PLCC | SRCC | PLCC | SRCC | PLCC | SRCC | PLCC | SRCC | PLCC |
| w/o MoAE | 149.9 | 3.5 | 0.765 | 0.792 | 0.858 | 0.885 | 0.927 | 0.918 | 0.852 | 0.898 | 0.800 | 0.866 | 0.750 | 0.768 | 0.681 | 0.672 |
| Last 4 layers | 231.8 | 7.5 | 0.830 | 0.859 | 0.933 | 0.944 | 0.954 | 0.952 | 0.937 | 0.960 | 0.866 | 0.909 | 0.853 | 0.867 | 0.735 | 0.732 |
| Last 6 layers | 272.7 | 10.2 | 0.851 | 0.871 | 0.940 | 0.949 | 0.957 | 0.952 | 0.949 | 0.966 | 0.870 | 0.910 | 0.863 | 0.878 | 0.740 | 0.737 |
| Last 8 layers | 313.6 | 13.4 | 0.852 | 0.883 | 0.941 | 0.947 | 0.956 | 0.951 | 0.953 | 0.967 | 0.872 | 0.913 | 0.866 | 0.875 | 0.746 | 0.743 |
| All 12 layers | 395.5 | 17.2 | 0.860 | 0.883 | 0.939 | 0.950 | 0.954 | 0.950 | 0.954 | 0.968 | 0.881 | 0.908 | 0.863 | 0.868 | 0.728 | 0.725 |

Table 7: Comparison with LIQE and UNIQUE when using the same training data.

| Dataset | LIVE | | CSIQ | | KADID | | BID | | LIVEC | | KonIQ | | Average | |
|---|---|---|---|---|---|---|---|---|---|---|---|---|---|---|
| Method | SRCC | PLCC | SRCC | PLCC | SRCC | PLCC | SRCC | PLCC | SRCC | PLCC | SRCC | PLCC | SRCC | PLCC |
| UNIQUE | 0.961 | 0.952 | 0.902 | 0.921 | 0.884 | 0.885 | 0.852 | 0.875 | 0.854 | 0.884 | 0.895 | 0.900 | 0.891 | 0.903 |
| LIQE | 0.970 | 0.951 | 0.936 | 0.939 | 0.930 | 0.931 | 0.875 | 0.900 | 0.904 | 0.910 | 0.919 | 0.908 | 0.922 | 0.923 |
| Gamma$^\dagger$ | 0.960 | 0.947 | 0.936 | 0.957 | 0.955 | 0.956 | 0.901 | 0.925 | 0.890 | 0.915 | 0.933 | 0.946 | **0.929** | **0.941** |

**Qualitative Results.** We visualize the image assessment results from different datasets, covering various scenarios, as shown in Figure 4. We can notice that our Gamma can accurately assess images from various tasks. These results shows the high generalization capability of our Gamma.

## 4.4 COMPARISON WITH OTHER MIXED TRAINING METHODS

In this subsection, we conduct a more detailed comparison with other mixed training methods. We first compare with LIQE and UNIQUE using the same training data and data splitting ratios. As shown in Table 7, our method achieves better performance on most datasets than LIQE and UNIQUE, especially on the KADID (+2.5% SRCC) and BID (+2.6% SRCC) datasets compared with LIQE. On other datasets, *i.e.*, LIVE and LIVEC, our model also achieves competitive results. Overall, our model has superior performance on these five datasets. In addition, we conduct cross dataset validation under this setting. As shown in Table 6, our method achieves highly competitive results on TID2013 and SPAQ, demonstrating the strong generalization capability of our method. Compared with Q-Align, as shown in Table 8, our method achieves better results on KonIQ and KADID, and is also highly competitive on SPAQ.

Table 6: Cross-dataset validation when using the same training data as LIQE and UNIQUE. The subscripts "$s$" and "$r$" stand for models trained on KADID and KonIQ, respectively.

| Dataset | TID2013 | SPAQ | AIGC2023 | Average |
|---|---|---|---|---|
| NIQE | 0.314 | 0.578 | - | 0.446 |
| DBCNN$_s$ | 0.686 | 0.412 | 0.730 | 0.609 |
| PaQ2PiQ | 0.423 | 0.823 | 0.643 | 0.630 |
| MUSIQ$_r$ | 0.584 | 0.853 | 0.736 | 0.724 |
| UNIQUE | 0.768 | 0.838 | 0.761 | 0.789 |
| LIQE | 0.811 | 0.881 | 0.744 | 0.812 |
| Gamma$^\dagger$ | 0.805 | 0.894 | 0.770 | **0.823** |

## 4.5 ABLATION STUDIES

We conduct detailed ablation studies to validate the effectiveness of our proposed modules. Note that we use naive prompt strategy (described in Section 3.2) to perform all ablations unless otherwise specified. We uniformly use 12 datasets for ablation experiments. Considering the page limit, we only show the datasets with relatively large differences in results.

Table 8: Comparison with Q-Align (Wu et al., 2023) when using the same training data.

| Dataset | KonIQ | | SPAQ | | KADID | |
|---|---|---|---|---|---|---|
| Method | SRCC | PLCC | SRCC | PLCC | SRCC | PLCC |
| Q-Align | 0.938 | 0.945 | **0.931** | **0.933** | 0.934 | 0.935 |
| Gamma$^\dagger$ | **0.940** | **0.950** | 0.928 | 0.932 | **0.962** | **0.964** |

**Effectiveness of the prompt strategy.** We propose the Scene-based Differential Prompt (SDP) to prompt different datasets. We evaluate the effectiveness of this strategy in Table 1. We can notice that the SDP strategy can improve the model performance on multiple datasets, especially on

Table 9: Sensitivity analysis of prompt. Quality prompt is {bad-quality, poor-quality, fair-quality, good-quality, perfect-quality}; General prompt replaces the scene prompt (detailed in Table 15) to "general", e.g., {underwear bad-quality image} to {general bad-quality image}.

| Dataset | LIVEC | | KonIQ | | LIVE | | CSIQ | | AGIQA3k | | UWIQA | | AVA | |
|---|---|---|---|---|---|---|---|---|---|---|---|---|---|---|
| Prompt | SRCC | PLCC | SRCC | PLCC | SRCC | PLCC | SRCC | PLCC | SRCC | PLCC | SRCC | PLCC | SRCC | PLCC |
| General prompt | 0.882 | 0.888 | 0.921 | 0.920 | 0.943 | 0.930 | 0.948 | 0.957 | 0.775 | 0.843 | 0.832 | 0.842 | 0.648 | 0.624 |
| Quality prompt | 0.885 | 0.889 | 0.931 | 0.940 | 0.950 | 0.946 | 0.946 | 0.951 | 0.822 | 0.872 | 0.861 | 0.876 | 0.451 | 0.455 |
| SDP | 0.891 | 0.914 | 0.939 | 0.950 | 0.953 | 0.953 | 0.960 | 0.968 | 0.887 | 0.923 | 0.873 | 0.884 | 0.750 | 0.749 |

Table 10: Results when only one adaptive expert is activated. The weights factors of other experts are set to 0. It can be observed that different experts focus on different datasets.

| Dataset | LIVEC | | KonIQ | | LIVE | | CSIQ | | AGIQA3k | | UWIQA | | GFIQA | | AVA | |
|---|---|---|---|---|---|---|---|---|---|---|---|---|---|---|---|---|
| Expert index | SRCC | PLCC | SRCC | PLCC | SRCC | PLCC | SRCC | PLCC | SRCC | PLCC | SRCC | PLCC | SRCC | PLCC | SRCC | PLCC |
| 1-th expert | **0.847** | **0.860** | **0.927** | **0.938** | **0.933** | **0.933** | **0.894** | **0.906** | 0.815 | 0.870 | **0.770** | **0.779** | **0.959** | **0.957** | 0.666 | 0.673 |
| 2-th expert | 0.715 | 0.672 | 0.681 | 0.717 | 0.900 | 0.861 | 0.815 | 0.846 | **0.832** | **0.885** | 0.755 | 0.756 | 0.826 | 0.797 | 0.663 | 0.652 |
| 3-th expert | 0.768 | 0.741 | 0.794 | 0.818 | 0.918 | 0.917 | 0.833 | 0.877 | 0.808 | 0.910 | 0.691 | 0.709 | 0.903 | 0.897 | **0.715** | **0.716** |
| Gamma | 0.851 | 0.871 | 0.940 | 0.949 | 0.957 | 0.952 | 0.949 | 0.966 | 0.870 | 0.910 | 0.863 | 0.878 | 0.970 | 0.970 | 0.740 | 0.737 |

CSIQ (+1.1% SRCC), LIVEC (+4% SRCC) and AGIQA-3k (+1.7 % SRCC). These results demonstrate that the SDP can effectively guide model learn differential features for different datasets, thus enhancing model performance. Furthermore, we ablate the SDP strategy and MOAE module respectively to explore their relationship and impact on model performance. As shown in Table 2, both methods can improve the performance of the model, such as +7.8% SRCC of SDP and +8.6% SRCC of MoAE on LIVEC. This shows the effectiveness of this adaptive expert feature learning and text guidance for multi-dataset learning. When the two methods are used simultaneously, the model can achieve the best results. Therefore, the two methods are mutually beneficial.

**The number of experts.** We explore the impact of different numbers of experts in the adaptive assessment experts. As shown in Table 3, the model achieves higher performance with more experts. This suggests that adding experts can better cope with the dataset bias problem when using a mixed training strategy. We use three experts to constitute the adaptive assessment experts in MoAE to achieve the optimal trade-off between accuracy and efficiency.

**Effect of freezing the shared expert.** We freeze the shared expert in the MoAE to retrain the multimodal capability of original model. This strategy also helps model capture the generalizable and common representation across varying contexts. Table 4 validates this method and shows that it is effective across various datasets.

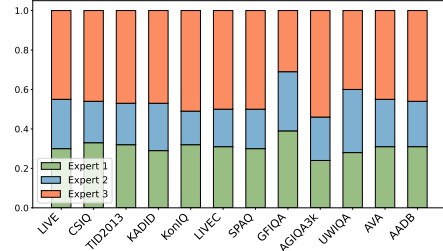

Figure 5: Average activations of three experts in the last layer of the visual encoder with naive prompts. Image evaluations of different scenes have different activation patterns.

**Merging features with factor $\sigma$.** Table 4 also demonstrates the effect of merging features of shared and adaptive experts with factor $\sigma$. We notice that this strategy improves the model performance on different datasets, especially on AVA and AGIQA3k. These results show that it is beneficial to utilize aligned features at the beginning of training and partially using features from adaptive experts.

**Adding adapter to last few layers.** We add the proposed MoAE module into the last few layers of the visual and text encoders. We compare the performance of adding MoAE to different numbers of layers in Table 5. After using MoAE, the performance of the model is significantly improved. We also observe that adding more than six layers of adapters does not improve model performance significantly, but further increases the model parameter and training overhead. Therefore, we integrate MoAE module in the last six layers of both visual and text encoder.

**Activation patterns of different datasets.** We visualize the average activation degree of three experts in the last layer of Gamma's visual encoder for different datasets, as shown in Figure 5. We can observe that the activation patterns are different for different scenarios. Specifically, the natural image assessment datasets, *e.g.*, LIVE, CSIQ, KADID, show different activation patterns from the

Table 11: Generalization capability validation on the exBeDDE and ECIQAD datasets. The "Pretrained weight" denotes the model weight of mixed training. We can notice that loading pretrained weight for initialization can improve model performance.

(a) Results on the exBeDDE datasets.

| Method | SRCC | PLCC |
|---|---|---|
| BRISQUE (Mittal et al., 2012a) | 0.890 | 0.906 |
| PSQA-I (Liu et al., 2019) | 0.907 | 0.924 |
| HyperIQA (Su et al., 2020) | 0.917 | 0.926 |
| FADE (Choi et al., 2015) | 0.714 | 0.729 |
| DHQI (Min et al., 2018) | 0.919 | 0.939 |
| VDA-DQA (Guan et al., 2022) | 0.923 | 0.942 |
| Ours | 0.916 | 0.938 |
| Ours + Pretrained weight | **0.937** | **0.951** |

(b) Results on the ECIQAD datasets.

| Method | SRCC | PLCC |
|---|---|---|
| BRISQUE (Mittal et al., 2012a) | 0.436 | 0.459 |
| BIQME (Gu et al., 2017) | 0.770 | 0.768 |
| BPRI (Min et al., 2017) | 0.152 | 0.181 |
| FRIQUEE (Ghadiyaram & Bovik, 2017) | 0.663 | 0.656 |
| CIQA (Chen et al., 2021) | 0.738 | 0.735 |
| ECIQ (Ke et al., 2021) | 0.839 | 0.842 |
| Ours | 0.912 | 0.922 |
| Ours + Pretrained weight | **0.917** | **0.927** |

face IQA dataset GFIQA and the underwater IQA dataset UWIQA. The synthetic distortion and authentic distortion dataset in nature IQA also have different activation patterns. These indicate that our MoAE module can assign experts with different activation levels to images of different scenarios, thereby capturing the discriminative features effectively.

**Sensitivity analysis of prompt.** We analyze the sensitivity of prompts when the model is trained with scene-based differential prompts (SDP). Table 9 shows that using prompts different from SDP slightly reduces performance on most datasets, showing the robustness of our method. The quality prompt performs better than the general prompt on the IQA task, but performs worse on the IAA task, indicating the importance of appropriate prompts. In conclusion, our method is robust and insensitive to prompts, nevertheless we suggest using correct prompts to obtain better performance.

**Analysis of the adaptive experts**. We add an experiment in which we only use one adaptive expert and set the router weights of the other experts to 0, to explore the preferences of different experts for different datasets. As shown in Table 10, the first expert performs well on most datasets, indicating it learns a general image assessment ability. The second and third experts focus on AIGC IQA and IAA tasks, respectively, and the third expert also shows excellent evaluation capabilities for natural images. These results indicate that different experts have learned domain-specific features of different datasets. They collaborate to achieve the powerful image assessment model Gamma.

### 4.6 GENERALIZATION CAPABILITY VALIDATION

We further validate the generalization capability of our method on two datasets, exBeDDE and ECIQAD. The exBeDDE is a dehazed IQA dataset and the ECIQAD is an enhanced colonoscopy IQA dataset, which belong to completely different evaluation domains compared to the used datasets in mixed training. We use naive prompt strategy for training and testing. The experimental results are reported in Table 11. We notice that our method can achieve competitive performance on these two datasets, showing the effectiveness and generalization capability of our method. More importantly, when we load the pretrained weight of Gamma for initialization, the performance of both datasets is improved and our method achieves the SOTA results. This indicates that our pretrained Gamma can be an effective foundation model to aid other assessment fields.

### 5 CONCLUSION

This paper introduces Gamma, a generic image assessment model that can be applied to various image scenarios. To achieve this, we utilize the mixed training of different datasets to obtain the assessment abilities of different scenarios. We propose a Mixture of Assessment Expert (MoAE) module and a Scene-based Differential Prompt (SDP) strategy to effectively cope with the MOS bias in different datasets. MoAE utilizes shared experts and adaptive experts to extract common and representative features adaptively. SDP strategy employs different prompts for different datasets to provide guidance for feature learning. Extensive experiments demonstrate that our method can achieve SOTA performance on various datasets simultaneously, showing the strong generalization and general image assessment capabilities.

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

## A   MORE IMPLEMENTATION DETAILS

### A.1   TRAINING DETAILS

We follow the typical training strategy to fine-tune each dataset, including random cropping and random horizontal flipping. We conduct all experiments on 3090 GPU. Mixed training of the 12 datasets takes 10 hours on a 3090 GPU. For the task-specific training, Table 12 shows the detailed training setting for the different datasets. We use the learning rate of 2e-5 for all datasets.

Table 12: Training settings for different datasets.

| Dataset | Task | Epoch | Batch size |
|---|---|---|---|
| LIVE (Sheikh et al., 2006) | SDN-IQA | 50 | 8 |
| CSIQ (Larson & Chandler, 2010) | SDN-IQA | 50 | 8 |
| TID2013 (Ponomarenko et al., 2013) | SDN-IQA | 20 | 8 |
| KADID (Lin et al., 2019) | SDN-IQA | 20 | 8 |
| CLIVE (Ghadiyaram & Bovik, 2015) | ADN-IQA | 50 | 8 |
| KonIQ (Hosu et al., 2020) | ADN-IQA | 20 | 8 |
| SPAQ (Fang et al., 2020) | ADN-IQA | 20 | 8 |
| GFIQA20k (Su et al., 2023b) | F-IQA | 10 | 8 |
| AGIQA3k (Li et al., 2023a) | AG-IQA | 20 | 8 |
| UWIQA (Yang et al., 2021a) | U-IQA | 50 | 8 |
| AVA (Murray et al., 2012) | IAA | 20 | 128 |
| AADB (Kong et al., 2016) | IAA | 20 | 8 |
| exBeDDE (Zhao et al., 2020) | D-IQA | 20 | 8 |
| ECIQAD (Yue et al., 2023) | EC-IQA | 20 | 8 |

### A.2   DATASETS

In this paper, we use a total of 14 datasets, 12 of which are used for unified training and 2 are used to evaluate the generalization ability of our model. We present the details of the used datasets in Table 13.

Table 13: Detail information about the 14 used datasets.

| Dataset | Task | Image Number | Label Type | Range |
|---|---|---|---|---|
| LIVE (Sheikh et al., 2006) | | 779 | DMOS | [1, 100] |
| CSIQ (Larson & Chandler, 2010) | SDN-IQA | 866 | DMOS | [0, 1] |
| TID2013 (Ponomarenko et al., 2013) | | 3,000 | MOS | [0, 9] |
| KADID-10k (Lin et al., 2019) | | 10,125 | MOS | [1, 5] |
| SPAQ (Fang et al., 2020) | | 11,125 | MOS | [0, 100] |
| LIVEC (Ghadiyaram & Bovik, 2015) | ADN-IQA | 1,162 | MOS | [1, 100] |
| KonIQ-10K (Hosu et al., 2020) | | 10,073 | MOS | [0, 100] |
| GFIQA20k (Su et al., 2023a) | F-IQA | 19,988 | MOS | [0, 1] |
| AGIQA3k (Li et al., 2023a) | AG-IQA | 2,982 | MOS | [0, 1] |
| UWIQA (Yang et al., 2021a) | U-IQA | 890 | MOS | [0, 1] |
| AVA (Murray et al., 2012) | IAA | 250,000 | MOS | [0, 10] |
| AADB (Kong et al., 2016) | IAA | 10,000 | MOS | [0, 1] |
| exBeDDE (Zhao et al., 2020) | D-IQA | 1670 | MOS | [0, 1] |
| ECIQAD (Yue et al., 2023) | EC-IQA | 2400 | MOS | [1, 9] |

### A.3   MODEL EFFICIENCY ANALYSIS

We calculate the number of parameters, computation, and inference time of our model. For inference time, we use a $224 \times 224$ image for testing. All indicators are obtained on a 3090 GPU. We compare it with two classic mixed training methods, LIQE (Zhang et al., 2023) and Q-Align (Wu et al., 2023). As shown in Table 14, our model achieves the best accuracy and efficiency. Compared with LIQE,

our model has significantly better performance. Compared with Q-Align, we not only have better performance, but also have significantly lower model parameters and inference latency.

Table 14: Detail information about the 14 used datasets.

| Method | Trainable Parms | FLOPs | Inference time | KonIQ SRCC | KADID SRCC |
|---|---|---|---|---|---|
| Q-Align (Wu et al., 2023) | 8.2B (8200M) | - | 0.1s | 0.938 | 0.934 |
| LIQE (Zhang et al., 2023) | 151M | 17.40G | 0.02s | 0.919 | 0.930 |
| Gamma | 122.8M | 28.45G | 0.025s | **0.939** | **0.962** |

## A.4 DETAILS OF THE SCENE-BASED DIFFERENTIAL PROMPT

In the Scene-based Differential Prompt, we use different prompts for datasets from different scene. Specifically, we divide datasets into five categories, *i.e.*, natural IQA, AI-generated IQA, underwater IQA, face IQA, natural IAA. We present the details in Table 15.

Table 15: Text prompts used in Scene-based Differential Prompt.

| Dataset | Prompt |
|---|---|
| LIVE, CSIQ, TID2013, KADID
LIVEC, KonIQ, SPAQ | `{natural bad-quality image`, `natural poor-quality image`,
`natural fair-quality image`,
`natural good-quality image`, `natural perfect-quality image}` |
| AGIQA3k | `{AI-generated bad-quality image`, `AI-generated poor-quality image`,
`AI-generated fair-quality image`,
`AI-generated good-quality image`, `AI-generated perfect-quality image}` |
| GFIQA20k | `{face bad-quality image`, `face poor-quality image`,
`face fair-quality image`,
`face good-quality image`, `face perfect-quality image}` |
| UWIQA | `{underwater bad-quality image`, `underwater poor-quality image`,
`underwater fair-quality image`,
`underwater good-quality image`, `underwater perfect-quality image}` |
| AVA, AADB | `{natural bad-aesthetics image`, `natural poor-aesthetics image`,
`natural fair-aesthetics image`,
`natural good-aesthetics image`, `natural perfect-aesthetics image}` |

