# OpenReview forum: "Gamma: Toward Generic Image Assessment with Mixture of Assessment Experts"
_ICLR.cc/2025/Conference — ICLR 2025 Conference Withdrawn Submission_

### Official Review · Reviewer_Gzj8 · 2024-11-02

**Soundness:** 3
**Presentation:** 3
**Contribution:** 2
**Rating:** 5
**Confidence:** 5

**Summary:**

This paper proposes a MOE approach for generic IQA approach. The main idea is to adopt a frozen CLIP as a general expert combined with trainable experts for different IQA tasks. The proposed approach demonstrate superiority on different IQA tasks compared with several existing baselines.

**Strengths:**

+ The motivation to combine multiple experts towards different IQA tasks is reasonable.
+ The paper is easy to follow

**Weaknesses:**

- The proposed approach still heavily relies on training data, though the bias can be alleviated by more data, but it may still fail in practical scenarios and training and testing on the data from the same sources cannot validate the generalization ability of the proposed approach, which is the key point that existing IQA approaches cannot overcome.
- The proposed approach lacks interpretation which is still another problem that existing approaches commonly have, i.e., what does each expert actually learn? Though with some reasonable designs, the proposed approach is still a black box.

**Questions:**

This paper has a reasonable motivation but fail to solve key problems existing in IQA from my opinion. My main concerns are as follows:

1. In Line 079, the author mentions that 'the primary challenge in mixed-dataset training is the mean opinion score (MOS) bias'. However, I do not find a reasonable solution on this challenge. It my understanding is correct, the author just try to adopt different experts and directly train these experts on the labeled data. This does not convince me on solving the bias across different datasets considering that the labels inherently have bias.

2. The proposed approach still heavily relies on training data, though the bias can be alleviated by more data, but it may still fail in practical scenarios. The proposed approach is  trained and tested on the same data sources, i.e., the 12 benchmarks mentioned in the paper. This cannot validate the generalization ability of the proposed approach, which is the key point that existing IQA approaches cannot overcome. The author may consider test on some other sources that have never been used during training for further validation.

3. The proposed approach lacks interpretation which is still another problem that existing approaches commonly have, i.e., what does each expert actually learn? Though with some reasonable designs, the proposed approach is still a black box. The author may consider explaining what each expert learns after training. Moreover, the improvement of 5 experts compared with 3  experts in Table 2 is very marginal and sometimes even worse. This is contradicted with the claim in Line 413. The author should provide more explanation.

4. There should be model complexity verification, including parameters, flops and inference time compared with other baselines.

5. I am a little confused about why a frozen CLIP can be directly adopted as a generic expert w/o finetuning on IQA datasets. Since it is never trained to do so. The author may provide a more detailed motivation on this.

---

> ### Author Response · Authors · 2024-11-25
> **Response to Reviewer Gzj8 [1/2]**
>
> Thanks for your professional and careful review. We respond to your concerns or questions as follows. We have modified the paper based on your valuable comments, marked in blue.
>
> > **Q1: The model may still fail in practical scenarios without validating the generalization ability of the proposed approach**
>
> - **Generalization ability validation**: It is very important to verify the generalization capability. Following the suggestion of reviewer F3iL, we perform cross dataset validation using the same data as LIQE and UNIQUE for training. As shown in the table below, our method achieves highly competitive results on TID2013 and SPAQ, demonstrating the strong generalization capability of our method. We supplement these discussions and experiments in Section 4.4 and Table 6, marked in blue.
>
> - **Practical usability**. Our model is trained on 12 datasets of different scenarios, so that it can directly cope with various evaluation scenarios. Due to its powerful image evaluation capabilities in multiple scenarios, it has many practical uses. For example, our model can be used as a data filtering tool to provide high-quality data for AIGC model training; it can also provide quality score guidance for image dehazing tasks[1]. In addition, the trained Gamma can be effectively applied as a pre-trained model to scenarios such as medical image evaluation (Section 4.6 and Table 8). In the future, we will collect more image evaluation data to further improve the generalization and versatility of the model.
>
> | Method          | TID2013 | SPAQ  | Average |
> |-----------------|---------|-------|---------|
> | **NIQE**        | 0.314   | 0.578 | 0.446   |
> | **DBCNN$_s$**   | 0.686   | 0.412 | 0.549   |
> | **PaQ2PiQ**     | 0.423   | 0.823 | 0.623   |
> | **MUSIQ$_r$**   | 0.584   | 0.853 | 0.719   |
> | **UNIQUE**      | 0.768   | 0.838 | 0.803   |
> | **LIQE**        | 0.811   | 0.881 | 0.846   |
> | **Gamma$^{+}$** | 0.804 | 0.893 | **0.849** |
>
> [1] Zhao S, Zhang L, Shen Y, et al. RefineDNet: A weakly supervised refinement framework for single image dehazing[J]. IEEE Transactions on Image Processing, 2021, 30: 3391-3404.
>
> ---
>
> > **Q2: The proposed approach lacks interpretation, i.e., what does each expert actually learn?.**
>
> - **Interpretation**. We conduct two analyses on the interpretability of the model. First, we calculate the average activation  level of experts under different datasets. As shown in Figure 5, image evaluations of different scenes have different activation patterns. This shows that the model has learned the differences (e.g., content differences and annotation differences) between datasets through the adaptive activation of experts.
> - **Interpretation**. In addition to the statistical analysis on the dataset, we add an experiment in which we only use one adaptive expert and set the router weights of the other experts to 0. In this way, we want to explore the preferences of different experts for different datasets. As shown in Table below, the first expert performs well on most datasets, indicating it learns a general image assessment ability. The second and third experts focus on AIGC IQA and IAA tasks, respectively, and the third expert also shows excellent evaluation capabilities for natural images. These results indicate that different experts have learned domain-specific features of different datasets. We add these analyses and results to Section 4.5 (Analysis of the adaptive experts) and Table 10.
>
> | Dataset       | LIVEC SRCC | LIVEC PLCC | KonIQ SRCC | KonIQ PLCC | LIVE SRCC | LIVE PLCC | CSIQ SRCC | CSIQ PLCC | AGIQA3k SRCC | AGIQA3k PLCC | UWIQA SRCC | UWIQA PLCC | GFIQA SRCC | GFIQA PLCC | AVA SRCC | AVA PLCC |
> |---------------|------------|------------|------------|------------|-----------|-----------|-----------|-----------|--------------|--------------|------------|------------|------------|------------|----------|----------|
> | **1-th expert** | **0.847**  | **0.860**  | **0.927**  | **0.938**  | **0.933** | **0.933** | **0.894** | **0.906** | 0.815        | 0.870        | **0.770**  | **0.779**  | **0.959**  | **0.957**  | 0.666    | 0.673    |
> | **2-th expert** | 0.715      | 0.672      | 0.681      | 0.717      | 0.900     | 0.861     | 0.815     | 0.846     | **0.832**    | **0.885**    | 0.755      | 0.756      | 0.826      | 0.797      | 0.663    | 0.652    |
> | **3-th expert** | 0.768      | 0.741      | 0.794      | 0.818      | 0.918     | 0.917     | 0.833     | 0.877     | 0.808        | 0.910        | 0.691      | 0.709      | 0.903      | 0.897      | **0.715**| **0.716**|
> | **Gamma**      | 0.851      | 0.871      | 0.940      | 0.949      | 0.957     | 0.952     | 0.949     | 0.966     | 0.870        | 0.910        | 0.863      | 0.878      | 0.970      | 0.970      | 0.740    | 0.737    |

---

> ### Author Response · Authors · 2024-11-25
> **Response to Reviewer Gzj8 [2/2]**
>
> >**Q3: How the model solve the bias across different datasets.**
>
> - **Problem Analysis**: We believe that when different datasets are mixed for training, their annotation biases make it difficult for the model to directly optimize for MOS. For example, a good quality image may have a high MOS in one dataset, but a low MOS in another dataset due to different annotation methods [1]. To address this issue, we propose the Mixture of Assessment Expert (MoAE) module, which can dynamically activate different experts to effectively learn the annotation patterns of different datasets. In addition, the proposed Scene-based Differential Prompt (SDP) strategy can also provide different meaningful features for different datasets, guiding the model to learn different representation.
> - **Experimental verification**: We conduct ablation experiments on the two methods, as shown in Table bellow. Due to the aforementioned problems, the model cannot achieve satisfactory performance without using MoAE and SDP. And two methods can improve the model performance effectively, proving that the two strategies can alleviate this problem well. We supplement these results and analysis in the Section 4.5 (Effectiveness of the prompt strategy) and Table 2.
> - **Phenomenon Analysis**: Moreover, we also show the activation level and performance of different experts for different datasets in Figure 5 and Table 10 (please see Q2 for details). We can observe that different experts have different activation patterns for different datasets (Figure 5) and different performances for different datasets (Table 10), which indicates that adaptive experts have learned the differences between different datasets.
>
> | MoAE  | SDP | LIVEC SRCC | LIVEC PLCC | KonIQ SRCC | KonIQ PLCC | LIVE SRCC | LIVE PLCC | CSIQ SRCC | CSIQ PLCC | AGIQA3k SRCC | AGIQA3k PLCC | UWIQA SRCC | UWIQA PLCC | AVA SRCC | AVA PLCC |
> |-----------------------|---------|------------|------------|------------|------------|-----------|-----------|-----------|-----------|--------------|--------------|------------|------------|----------|----------|
> | ✗   | ✗|0.765      | 0.792      | 0.858      | 0.885      | 0.927     | 0.918     | 0.852     | 0.898     | 0.800        | 0.866        | 0.750      | 0.768      | 0.681    | 0.672    |
> | ✗   | ✓|0.843      | 0.856      | 0.874      | 0.896      | 0.929     | 0.917     | 0.866     | 0.901     | 0.841        | 0.887        | 0.770      | 0.780      | 0.721    | 0.715    |
> | ✓   | ✗ |0.851      | 0.871      | 0.940      | 0.949      | 0.957     | 0.952     | 0.949     | 0.966     | 0.870        | 0.910        | 0.863      | 0.878      | 0.740    | 0.737    |
> | ✓   | ✓|0.891      | 0.914      | 0.939      | 0.950      | 0.960     | 0.968     | 0.953     | 0.953     | 0.887        | 0.923        | 0.873      | 0.884      | 0.750    | 0.749    |
>
> [1] Zhang W, Ma K, Zhai G, et al. Uncertainty-aware blind image quality assessment in the laboratory and wild[J]. IEEE Transactions on Image Processing, 2021, 30: 3474-3486.
>
> ---
>
> > **Q4: There should be model complexity verification.**
> Thank you for your suggestion. We calculate the number of parameters, FLOPs, and inference time of our model. We compare 2 classical mixed training methods, i.e., the CLIP-based method LIQE and the large language model method Q-Align. As shown in the table below, our model achieves the best accuracy and efficiency. Compared with LIQE, our model has significantly better performance. Compared with Q-Align, we not only have better performance, but also have significantly lower model parameters and inference latency. Therefore, our model is a better choice when serving as an effective tool in other fields. We have added this part to Appendix A.3 and Table 14, marked it in blue.
>
> | Method |Trainable Parms | FLOPs | Inference time | KonIQ SRCC | KADID SRCC |
> | --- | --- | --- | --- | --- | --- |
> | Q-Align |  8.2B (8200M) | -  | 0.1s|0.938|0.934|
> | LIQE |  151M| 17.4G | 0.02s|0.919|0.930|
> | Ours | 122.8M | 28.45G |0.025s|**0.939**| **0.962**|
>
> ---
>
> > **Q5: Why a frozen CLIP can be directly adopted as a generic expert w/o finetuning on IQA datasets.**
>
> We would like to clarify that we use the pre-trained model UniQA instead of CLIP as our pre-trained weights. We introduce our pre-trained architecture and weights in Section 3.1. UniQA is pre-trained on large-scale quality and aesthetics related image and text data, so it can be used as a general expert. However, UniQA needs to be fine-tuned to be applied to specific tasks and cannot evaluate multiple image scenarios at the same time. To address the content diversity and label bias issues of mixed-datasets training,  we propose Mixture of Assessment Experts (MoAE) and Scene-based Differential Prompt (SDP), to build a general image evaluation model Gamma.

---

> ### Comment · Reviewer_Gzj8 · 2024-11-26
>
> The author addressed most of my concerns.
>
> The main concern still remains for Q1, the author should also provide the test result on the two training set for comparison with the results on cross datasets. It seems that the results on cross data which the model has not seen during training still decreases. This is actually the problem that all other IQA methods have.
>
> Also, the advantage of the proposed approach compared with LIQE is not obvious, this may indicate the proposed MOE for IQA may not increase the robustness of the proposed approach across dataset.
>
> I should have increased my score but I agree with Reviewer F3iL that unfair experiment comparison without detailed explanation should be avoided in academic paper. I hope the author should correct this and make fair and solid experiments in future submissions.

---

> ### Author Response · Authors · 2024-11-28
> **Response to Reviewer Gzj8**
>
> Thank you for your valuable further response. To your concerns, we have responded as follows. If you have any other questions, we would be more than happy to respond!
>
> > **Q1: The main concern still remains for Q1, the author should also provide the test result on the two training set for comparison with the results on cross datasets. It seems that the results on cross data which the model has not seen during training still decreases. This is actually the problem that all other IQA methods have.**
>
> Since different datasets have different testing standards, direct zero-shot testing often cannot achieve performance similar to fine-tuning. Our work attempts to achieve better generalization performance by combining more datasets. We proposed Mixture of Assessment Expert (MoAE) and Scene-based Differential Prompt (SDP) to solve the problems of labeling bias and content diversity for multi-dataset training. As shown in the table below, zero-shot performance will decrease compared to task-specific fine-training. **However, it is worth noting that Gamma$^{++}$, which is trained on 12 datasets, achieves significant improvements on the AIGC2023 dataset with zero-shot testing**. It achieves 0.818 SRCC, an improvement of 7.4% SRCC than LIQE and 4.8% SRCC than Gamma$^{+}$. This demonstrates the benefits of unified training and the feasibility of making a robust and general image assessment model. In the future, we will use our framework to train on more data to achieve better cross dataset performance.
>
> **Cross dataset results.** **Gamma$^{+}$** uses 6 datasets for training and performs zero-shot Cross dataset validation on other datasets.  **Gamma$^{++}$** uses 12 datasets (include TID2012 and SPAQ) for training and performs zero-shot Cross dataset validation on other datasets. **Gamma$^{*}$** uses task-specific fine-tuning on these datasets.
>
> | Method          | TID2013 | SPAQ  | AIGC2023 |
> |-----------------|---------|-------|----------|
> | **NIQE**        | 0.314   | 0.578 | -        |
> | **DBCNN$_s$**   | 0.686   | 0.412 | 0.730    |
> | **PaQ2PiQ**     | 0.423   | 0.823 | 0.643    |
> | **MUSIQ$_r$**   | 0.584   | 0.853 | 0.736    |
> | **UNIQUE**      | 0.768   | 0.838 | 0.761    |
> | **LIQE**        | 0.811   | 0.881 | **0.744**    |
> | **Gamma$^{+}$** | 0.805 | 0.894 | 0.770    |
> | **Gamma$^{++}$** | - | - | **0.818**    | - |
> | **Gamma$^{*}$** | 0.944 | 0.950 | 0.862    |
>
> ---
>
> > **Q2: Also, the advantage of the proposed approach compared with LIQE is not obvious, this may indicate the proposed MOE for IQA may not increase the robustness of the proposed approach across dataset.**
>
> We add the BID dataset and retrain our model using the same 7:1:2 ratio as LIQE. The results are shown in the table below.  We can observe that our method has highly competitive results, especially on the KADID, BID, and KonIQ datasets. Our method has the best average performance, achieving an average of 0.929 SRCC (vs. 0.922 of LIQE) and 0.941 SRCC (vs. 0.923 of LIQE)
>
> | Method         | LIVE SRCC | LIVE PLCC | CSIQ SRCC | CSIQ PLCC | KADID SRCC | KADID PLCC | BID SRCC | BID PLCC | LIVEC SRCC | LIVEC PLCC | KonIQ SRCC | KonIQ PLCC | Average SRCC | Average PLCC |
> |----------------|-------------|-------------|-------------|-------------|--------------|--------------|------------|------------|--------------|--------------|--------------|--------------|----------------|----------------|
> | **UNIQUE**     | 0.961       | 0.952       | 0.902       | 0.921       | 0.884        | 0.885        | 0.852      | 0.875      | 0.854        | 0.884        | 0.895        | 0.900        | 0.891          | 0.903          |
> | **LIQE**       | 0.970       | 0.951       | 0.936       | 0.939       | 0.930        | 0.931        | 0.875      | 0.900      | 0.904        | 0.910        | 0.919        | 0.908        | 0.922          | 0.923          |
> | **Gamma$^{+}$** | 0.960    | 0.947       | 0.936       | 0.957       | 0.955        | 0.956        | 0.901      | 0.925      | 0.890        | 0.915        | 0.933        | 0.946        | **0.929**      | **0.941**      |
>
>
> > **Q3: unfair experiment comparison.**
>
> Following Reviewer F3iL's suggestion, we have retrained our model using the same data and data splitting ratio as LIQE and UNIQUE to ensure the fairness of the experiment. Please refer to the table in Q2 for the results. We can observe that we have obviously superior overall performance, which proves the superiority of our method.

---

> ### Author Response · Authors · 2024-12-01
> **To Reviewer Gzj8**
>
> Dear Reviewer Gzj8,
>
> We truly appreciate your guidance to advance our work. We genuinely value the time and effort you dedicated to reviewing our paper. Considering that the discussion will end soon, we eagerly look forward to your response.
>
> Best regards,
>
> Authors

---

> > ### Author Response · Authors · 2024-12-03
> > **To Reviewer Gzj8**
> >
> > Dear Reviewer Gzj8,
> >
> > We truly appreciate your guidance to advance our work. We genuinely value the time and effort you dedicated to reviewing our paper. Considering that the discussion will end soon, we eagerly look forward to your response.
> >
> > Best regards,
> >
> > Authors

---

> > > ### Author Response · Authors · 2024-12-03
> > > **To Reviewer Gzj8**
> > >
> > > Dear Reviewer Gzj8,
> > >
> > > We truly appreciate your guidance to advance our work. We genuinely value the time and effort you dedicated to reviewing our paper. Considering that the discussion will end soon, we eagerly look forward to your response.
> > >
> > > We will open source this large-scale pre-trained image evaluation model. It has excellent image evaluation capabilities in a variety of scenarios. Our model has broad applications in various real-world scenarios, such as selecting high-quality images in a data engine, or acting as reward models when aligning image generative models with human feedback. In addition, it can be used as a base model to assist other downstream tasks, such as medical image quality assessment (Table 11b).
> > >
> > > Best regards,
> > >
> > > Authors

---

### Official Review · Reviewer_jhUC · 2024-11-03

**Soundness:** 3
**Presentation:** 3
**Contribution:** 2
**Rating:** 6
**Confidence:** 4

**Summary:**

This paper proposes a generic image evaluation model, Gamma, based on hybrid evaluation experts, which is trained to efficiently evaluate images from different scenes through mixed data sets. Taking into account annotation bias of different datasets, the authors propose a hybrid evaluation expert module that uses shared and adaptive experts to dynamically learn common and specific knowledge of different datasets respectively. At the same time, a scenario-based differential cue strategy is introduced to enhance the adaptability to various scenarios. They conducted an empirical study on 12 data sets and compared it with the existing models, and the results reached the most advanced level at present.

**Strengths:**

The author can take into account the annotation bias of different data sets and creatively propose a hybrid evaluation expert module. This paper will help implement a unified evaluation standard on different data sets in the future. This achievement is commendable. Experiments conducted by the authors on multiple datasets effectively demonstrate the universality of their model, and the introduction of the scenario-based differential cue strategy also proves its effectiveness through these experiments.

**Weaknesses:**

The article itself is to solve the annotation bias of different data sets, but by adding data, there is a suspicion of writing answers for people to change the question. The personalized tendencies among the hired experts are not addressed, and it is difficult to say that the results of the hired experts' data tuning are closer to the real situation than the previous results
At the same time, the author chooses to adjust the model only in the rear module instead of all modules, and says that this method reduces the computing power requirements of the model, which is obvious, but for the reason why this choice is made, does the benefit in terms of reducing the computing power really outweigh the model quality? Is this really worth it? The author doesn't offer a convincing explanation.

**Questions:**

1. I hope the author can prove that the results after data tuning by the hired experts are closer to the real situation than the previous results.
2. I would like the author to explain the basis on which the model was chosen.
3. I hope that the author can add an experimental result according to the theory of the model in the paper without referring to the data of several hired experts.
4. I'm concerned about how SCENE-BASED DIFFERENTIAL PROMPT is implemented?

---

> ### Author Response · Authors · 2024-11-25
> **Response to Reviewer jhUC [1/3]**
>
> Thanks for your professional and careful review. We respond to your concerns or questions as follows. We have modified the paper based on your valuable comments, marked in blue.
>
>
> > **Q1: The authors solve the annotation bias of different data sets by adding data, which is a suspicion of writing answers for people to change the question.**
>
> We would like to clarify that in order to achieve a more general image assessment model with strong generalization performance, we need to collect datasets of various scenes for training. However, the bias of labels and the diversity of content bring challenges to the training of the model. Therefore, we propose a mixture of experts module and a scene-based prompt strategy to improve the model's ability to handle different datasets. We demonstrate that both of our methods can help improve model performance through ablation experiments (see Table below in the Q2 section).
>
> ---
>
> > **Q2: Prove that the results after data tuning by the hired experts are closer to the real situation than the previous results.**
>
> - **Problem Analysis**: We believe that when different datasets are mixed for training, their annotation biases make it difficult for the model to directly optimize for MOS. For example, a good quality image may have a high MOS in one dataset, but a low MOS in another dataset due to different annotation methods [1]. To address this issue, we propose the Mixture of Assessment Expert (MoAE) module, which can dynamically activate different experts to effectively learn the annotation patterns of different datasets. In addition, the proposed Scene-based Differential Prompt (SDP) strategy can also provide different meaningful features for different datasets, guiding the model to learn different representation.
> - **Experimental verification**: We conduct ablation experiments on the two methods, as shown in Table bellow. Due to the aforementioned problems, the model cannot achieve satisfactory performance without using MoAE and SDP. And two methods can improve the model performance effectively, proving that the two strategies can alleviate this problem well. We supplement these results and analysis in the Section 4.5 (Effectiveness of the prompt strategy) and Table 2.
> - **Phenomenon Analysis**: Moreover, we also show the activation level and performance of different experts for different datasets in Figure 5 and Table 10 (please see Q4 for details). We can observe that different experts have different activation patterns for different datasets (Figure 5) and different performances for different datasets (Table 10), which indicates that adaptive experts have learned the differences between different datasets.
>
> | MoAE  | SDP | LIVEC SRCC | LIVEC PLCC | KonIQ SRCC | KonIQ PLCC | LIVE SRCC | LIVE PLCC | CSIQ SRCC | CSIQ PLCC | AGIQA3k SRCC | AGIQA3k PLCC | UWIQA SRCC | UWIQA PLCC | AVA SRCC | AVA PLCC |
> |-----------------------|---------|------------|------------|------------|------------|-----------|-----------|-----------|-----------|--------------|--------------|------------|------------|----------|----------|
> | ✗   | ✗|0.765      | 0.792      | 0.858      | 0.885      | 0.927     | 0.918     | 0.852     | 0.898     | 0.800        | 0.866        | 0.750      | 0.768      | 0.681    | 0.672    |
> | ✗   | ✓|0.843      | 0.856      | 0.874      | 0.896      | 0.929     | 0.917     | 0.866     | 0.901     | 0.841        | 0.887        | 0.770      | 0.780      | 0.721    | 0.715    |
> | ✓   | ✗ |0.851      | 0.871      | 0.940      | 0.949      | 0.957     | 0.952     | 0.949     | 0.966     | 0.870        | 0.910        | 0.863      | 0.878      | 0.740    | 0.737    |
> | ✓   | ✓|0.891      | 0.914      | 0.939      | 0.950      | 0.960     | 0.968     | 0.953     | 0.953     | 0.887        | 0.923        | 0.873      | 0.884      | 0.750    | 0.749    |
>
> [1] Zhang W, Ma K, Zhai G, et al. Uncertainty-aware blind image quality assessment in the laboratory and wild[J]. IEEE Transactions on Image Processing, 2021, 30: 3474-3486.

---

> ### Author Response · Authors · 2024-11-25
> **Response to Reviewer jhUC [2/3]**
>
> > **Q3: Why the authors only add MoAE to the last few layers of the model? Explain the basis on which the model was chosen.**
>
> We choose to use six-layer MoAE to the last few layers of the model to achieves the best trade-off between accuracy and efficiency. We have added training time and model parameter quantity indicators in Table 5 for comparison. As shown in Table 5 (below), adding the MoAE module can significantly improve the performance of the model. We observe that when the MoAE module adds more than 6 layers, the performance of the model will not be significantly improved, but the model parameters and training cost will be further increased. Therefore, we choose to use six-layer MoAE. We supplement these discussions and results in Section 4.5 (Adding adapter to last few layers) and Table 5, marked in blue.
>
> | MoE Layer        | Parms (Million) | FLOPs (Hours) | LIVEC SRCC | LIVEC PLCC | KonIQ SRCC | KonIQ PLCC | LIVE SRCC | LIVE PLCC | CSIQ SRCC | CSIQ PLCC | AGIQA3k SRCC | AGIQA3k PLCC | UWIQA SRCC | UWIQA PLCC | AVA SRCC | AVA PLCC |
> |------------------|-----------------|---------------|------------|------------|------------|------------|-----------|-----------|-----------|-----------|--------------|--------------|------------|------------|----------|----------|
> | w/o MoAE         | 149.9          | 3.5           | 0.765      | 0.792      | 0.858      | 0.885      | 0.927     | 0.918     | 0.852     | 0.898     | 0.800        | 0.866        | 0.750      | 0.768      | 0.681    | 0.672    |
> | Last 4 layers    | 231.8          | 7.5           | 0.830      | 0.859      | 0.933      | 0.944      | 0.954     | 0.952     | 0.937     | 0.960     | 0.866        | 0.909        | 0.853      | 0.867      | 0.735    | 0.732    |
> | **Last 6 layers** | **272.7**      | **10.2**     | 0.851  | 0.871  | 0.940  | 0.949  | 0.957 | 0.952 | 0.949 | 0.966 | 0.870    | 0.910    | 0.863  | 0.878  | 0.740| 0.737|
> | Last 8 layers    | 313.6          | 13.4          | 0.852      | 0.883      | 0.941      | 0.947      | 0.956     | 0.951     | 0.953     | 0.967     | 0.872        | 0.913        | 0.866      | 0.875      | 0.746    | 0.743    |
> | All 12 layers    | 395.5          | 17.2          | 0.860      | 0.883      | 0.939      | 0.950      | 0.954     | 0.950     | 0.954     | 0.968     | 0.881        | 0.908        | 0.863      | 0.868      | 0.728    | 0.725    |

---

> ### Author Response · Authors · 2024-11-25
> **Response to Reviewer jhUC [3/3]**
>
> > **Q4: Adding an experimental result according to the theory of the model in the paper without referring to the data of several hired experts.**
> - **Experimental results:** We add an experiment in which we only use one adaptive expert and set the router weights of the other experts to 0. In this way, we want to explore the preferences of different experts for different datasets. As shown in Table below, the first expert performs well on most datasets, indicating it learns a general image assessment ability. The second and third experts focus on AIGC IQA and IAA tasks, respectively, and the third expert also shows excellent evaluation capabilities for natural images. These results indicate that different experts have learned domain-specific features of different datasets. We added these analyses and results to Section 4.5 (Analysis of the adaptive experts) and Table 10.
> - **For the number of experts**: The number of experts is related to the overall distribution of the data. We found that using 3 experts achieves excellent results and adding more experts does not significantly improve performance through ablation experiments. In addition, in other areas such as large language models [1-2], the number of experts in the MoE structure is also usually set empirically.
>
> | Dataset       | LIVEC SRCC | LIVEC PLCC | KonIQ SRCC | KonIQ PLCC | LIVE SRCC | LIVE PLCC | CSIQ SRCC | CSIQ PLCC | AGIQA3k SRCC | AGIQA3k PLCC | UWIQA SRCC | UWIQA PLCC | GFIQA SRCC | GFIQA PLCC | AVA SRCC | AVA PLCC |
> |---------------|------------|------------|------------|------------|-----------|-----------|-----------|-----------|--------------|--------------|------------|------------|------------|------------|----------|----------|
> | **1-th expert** | **0.847**  | **0.860**  | **0.927**  | **0.938**  | **0.933** | **0.933** | **0.894** | **0.906** | 0.815        | 0.870        | **0.770**  | **0.779**  | **0.959**  | **0.957**  | 0.666    | 0.673    |
> | **2-th expert** | 0.715      | 0.672      | 0.681      | 0.717      | 0.900     | 0.861     | 0.815     | 0.846     | **0.832**    | **0.885**    | 0.755      | 0.756      | 0.826      | 0.797      | 0.663    | 0.652    |
> | **3-th expert** | 0.768      | 0.741      | 0.794      | 0.818      | 0.918     | 0.917     | 0.833     | 0.877     | 0.808        | 0.910        | 0.691      | 0.709      | 0.903      | 0.897      | **0.715**| **0.716**|
> | **Gamma**      | 0.851      | 0.871      | 0.940      | 0.949      | 0.957     | 0.952     | 0.949     | 0.966     | 0.870        | 0.910        | 0.863      | 0.878      | 0.970      | 0.970      | 0.740    | 0.737    |
>
> [1] Dai D, Deng C, Zhao C, et al. Deepseekmoe: Towards ultimate expert specialization in mixture-of-experts language models[J]. arXiv preprint arXiv:2401.06066, 2024.
>
> [2] Jiang A Q, Sablayrolles A, Roux A, et al. Mixtral of experts[J]. arXiv preprint arXiv:2401.04088, 2024.
>
> ---
>
> > **Q5: How SCENE-BASED DIFFERENTIAL PROMPT is implemented?**
>
> - In implementation, we will assign the corresponding prompt according to the image path of the input image. For example, image from KonIQ dataset with "koniq/xxx.jpg" will be assigned as natural IQA image, i.e., the prompt is {natural bad-quality image, natural poor-quality image, natural fair-quality image, natural good-quality image, natural perfect-quality image}.
> - If the user knows the scene type of the image, e.g., natural or AIGC image, they can use a specific prompt for inference (Gamma$^{+}$). If the user does not know the scene type of the image, they can use the model (Gamma) that is not trained using SCENE-BASED DIFFERENTIAL PROMPT, which can also achieve excellent image assessment performance across 12 datasets.

---

> ### Author Response · Authors · 2024-11-28
> **Response to Reviewer jhUC**
>
> Thanks for raising the score. We truly appreciate your recognition of our work, which encourages our further work. Best wishes.

---

### Official Review · Reviewer_F3iL · 2024-11-03

**Soundness:** 2
**Presentation:** 3
**Contribution:** 2
**Rating:** 5
**Confidence:** 5

**Summary:**

This manuscript introduces a mixture of assessment experts and scene-based prompts to achieve high-performing, unified image quality and aesthetics assessment across diverse datasets and scenarios.

**Strengths:**

1. The idea of using MoAE to overcome the dataset distribution gap is reasonable.
2. The performance is better than baselines (but may be unfair, see Weaknesses).
3. The paper is presented clearly and easy to follow.

**Weaknesses:**

1. **The comparison with baseline methods is unfair**. Table 1 contains some blanks for baseline methods like UNIQUE and LIQE, which raises concerns about the experimental setup. I have carefully checked the results of UNIQUE and LIQE and ensured that these numbers are directly copied from previous papers. The training datasets of UNIQUE and LIQE differ from this manuscript, which is unfair.
2. **The generalization experiments are not enough**. Though this manuscript mentions that 12 datasets are involved, most of them are used in training. Only two of them are used to evaluate the generalization ability. More results from cross-dataset experiments are needed. For example, for the seven datasets in Table 1, how about the results of training with three datasets and evaluating with another four datasets?
3. **The manuscript does not compare with an important baseline**, Q-Align, which also proposes a unified framework that can co-train multiple datasets. Moreover, only training on three datasets, Q-Align’s results on some datasets have surpassed this manuscript.
4. **There is no analysis of efficiency**, though this manuscript claims the proposed method is both effective and efficient.  Please report the comparison of the number of parameters, FLOPs, and training / inference time to support the claim.
5. **There is no sensitivity analysis of prompts**. This manuscript uses scene-based differential prompts to improve the ability across multiple datasets and scenes. However, it is risky that the model will be highly reliant on such prompts. During testing, if the prompts are changed, the performance may significantly drop. Therefore, a detailed analysis of the sensitivity to prompts should be included.

**Questions:**

See Weaknesses.

---

> ### Author Response · Authors · 2024-11-25
> **Response to Reviewer F3iL [1/2]**
>
> We sincerely appreciate your helpful feedback. Your guidance is crucial in advancing our work. We have modified the paper based on your valuable comments, marked in blue.
>
> > **Q1: The comparison with baseline methods is unfair.**
>
> For a fair comparison, we use the same training data as UNIQUE and LIQE. As shown in the Table below, our method achieves better performance on most datasets, especially on the KADID (+2.5% SRCC) and KonIQ (+1.5% SRCC) datasets compared with LIQE. On other datasets, i.e., LIVE and LIVEC, our model also achieves competitive results. Overall, our model has superior performance on these five datasets. We present these results in Section 4.4 and Table 7, marked in blue.
>
> | Method         | LIVE SRCC | LIVE PLCC | CSIQ SRCC | CSIQ PLCC | KADID SRCC | KADID PLCC | LIVEC SRCC | LIVEC PLCC | KonIQ SRCC | KonIQ PLCC | Average SRCC | Average PLCC |
> |----------------|-------------|-------------|-------------|-------------|--------------|--------------|--------------|--------------|--------------|--------------|----------------|----------------|
> | **UNIQUE**     | 0.969       | 0.968       | 0.902       | 0.927       | 0.878        | 0.876        | 0.854        | 0.890        | 0.896        | 0.901        | 0.900          | 0.912          |
> | **LIQE**       | 0.970       | 0.951       | 0.936       | 0.939       | 0.930        | 0.931        | 0.904        | 0.910        | 0.919        | 0.908        | 0.932          | 0.928          |
> | **Gamma$^{+}$** | 0.965   | 0.953       | 0.938       | 0.951       | 0.955        | 0.957        | 0.882        | 0.896        | 0.934        | 0.945        | **0.935**      | **0.940**      |
>
> ---
>
> > **Q2: The generalization experiments are not enough.**
>
> We perform cross dataset validation using the same data as LIQE and UNIQUE for training. As shown in Table below, our method achieves highly competitive results on TID2013 and SPAQ, demonstrating the strong generalization capability of our method. We present these results in Section 4.4 and Table 6, marked in blue.
>
> The subscripts $s$ and $r$ stand for models trained on KADID and KonIQ, respectively.
>
> | Method          | TID2013 | SPAQ  | Average |
> |-----------------|---------|-------|---------|
> | **NIQE**        | 0.314   | 0.578 | 0.446   |
> | **DBCNN$_s$**   | 0.686   | 0.412 | 0.549   |
> | **PaQ2PiQ**     | 0.423   | 0.823 | 0.623   |
> | **MUSIQ$_r$**   | 0.584   | 0.853 | 0.719   |
> | **UNIQUE**      | 0.768   | 0.838 | 0.803   |
> | **LIQE**        | 0.811   | 0.881 | 0.846   |
> | **Gamma$^{+}$** | 0.804 | 0.893 | **0.849** |
>
> ---
>
> > **Q3: The manuscript does not compare with an important baseline.**
>
> We compare Q-Align fairly with the same training data. As shown in the table below, our method achieves better results on KonIQ and KADID, and is also highly competitive on SPAQ. Compared with Q-Align, our model is more efficient. Q-Align uses a heavy language model with 8.2B (8200M) parameters, while our Gamma has only 122.8M trainable parameters (272.7M in total). Therefore, our model is more efficient and effective than Q-Align. In practical scenarios, such as using IQA models for AIGC data filtering, our approach is more resource-friendly. We supplement these discussions and results in Section 4.4 and Table 8, marked in blue.
>
> | Dataset | KonIQ SRCC | KonIQ PLCC | SPAQ SRCC | SPAQ PLCC | KADID SRCC | KADID PLCC |
> |---------|------------|------------|-----------|-----------|------------|------------|
> | Q-Align | 0.938      | 0.945      | **0.931** | **0.933** | 0.934      | 0.935      |
> | **Gamma$^{+}$** | **0.940**   | **0.950**   | 0.928     | 0.932     | **0.962**   | **0.964**   |

---

> ### Author Response · Authors · 2024-11-25
> **Response to Reviewer F3iL [2/2]**
>
> > **Q4: There is no analysis of efficiency.**
>
> - **Our work is not focused on lightweight model design.** Firstly, we would like to clarify that our work is not focused on lightweight model design. We freeze most of the parameter of CLIP and only train the adapter and adaptive experts modules (only add to the last six layers of model), which is a parameter-efficient fine-tuning method. This method is more efficient than the previous method of directly training CLIP models such as LIQE.
> - **Comparison of efficiency with other baseline methods.** We compare LIQE, Q-Align and our method in terms of model parameter, FLOPs and inference speed. As shown in the table below, our model achieves the best accuracy and efficiency. Compared with LIQE, our model has significantly better performance. Compared with Q-Align, we not only have better performance, but also have significantly lower model parameters and inference latency. Therefore, our model is a better choice when serving as an effective tool in other fields. We have added this part to Appendix A.3 and Table 14 and marked it in blue.
>
> | Method |Trainable Parms | FLOPs | Inference time | KonIQ SRCC | KADID SRCC |
> | --- | --- | --- | --- | --- | --- |
> | Q-Align |  8.2B (8200M) | -  | 0.1s|0.938|0.934|
> | LIQE |  151M| 17.4G | 0.02s|0.919|0.930|
> | Ours | 122.8M | 28.45G |0.025s|**0.939**| **0.962**|
>
> ---
>
> > **Q5: There is no sensitivity analysis of prompts.**
>
> According to your valuable suggestion, we add a sensitivity analysis of prompt. We test two types of other prompt, General prompt and Quality prompt. General prompt replaces the scene prompt to “general”, e.g.,  *{underwater bad-quality image}* to *{general bad-quality image}*, thus the general prompt is *{general bad-quality image, general poor-quality image, general fair-quality image, general good-quality image, general perfect-quality image}*;  Quality prompt is *{bad-quality image, poor-quality image, fair-quality image, good-quality image, perfect-quality image}*. As shown in the table below, we can observe that using  prompts different from SDP slightly reduces performance on most datasets, showing the robustness of our approach. The quality prompt performs better than the general prompt on the IQA task, but performs worse on the IAA task, indicating the importance of appropriate prompts. In conclusion, our method is robust and insensitive to prompts, nevertheless we suggest using correct prompts to obtain better performance. We supplement these discussions and results in Section 4.5 (Sensitivity analysis of prompt) and Table 9, marked in blue.
>
> | Prompt           | LIVEC SRCC | LIVEC PLCC | KonIQ SRCC | KonIQ PLCC | LIVE SRCC | LIVE PLCC | CSIQ SRCC | CSIQ PLCC | AGIQA3k SRCC | AGIQA3k PLCC | UWIQA SRCC | UWIQA PLCC | AVA SRCC | AVA PLCC |
> |------------------|--------------|--------------|--------------|--------------|-------------|-------------|-------------|-------------|----------------|----------------|--------------|--------------|------------|------------|
> | **General prompt** | 0.882      | 0.888        | 0.921        | 0.920        | 0.943       | 0.930       | 0.948       | 0.957       | 0.775          | 0.843          | 0.832        | 0.842        | 0.648      | 0.624      |
> | **Quality prompt** | 0.885      | 0.889        | 0.931        | 0.940        | 0.950       | 0.946       | 0.946       | 0.951       | 0.822          | 0.872          | 0.861        | 0.876        | 0.451      | 0.455      |
> | **SDP**          | 0.891       | 0.914        | 0.939        | 0.950        | 0.953       | 0.953       | 0.960       | 0.968       | 0.887          | 0.923          | 0.873        | 0.884        | 0.750      | 0.749      |

---

> > ### Comment · Reviewer_F3iL · 2024-11-26
> > **Official Comment by Reviewer F3iL**
> >
> > Thanks for the authors' detailed responses.
> >
> > - My concerns about Q4 and Q5 are well solved.
> >
> > - Small concern in Q2 and Q3. The advantage of this work is not significant.
> >
> > - Main concern in Q1.
> >
> >   - First, LIQE and UNIQUE all include the BID dataset, which is missing here.
> >   - Second, the training/test splits in UNIQUE and LIQE are different. UNIQUE uses 80% training and 20% testing. LIQE  takes 70% training, 10% validation, and 20% testing. Directly copying their results is not right. Please refer to LIQE for the right way to compare. LIQE also compares with UNIQUE, but it re-trains UNIQUE under the same splits.
> >   - Finally, I think that **unfair comparison, especially in the main table of the manuscript, can only be solved by resubmission**.
> >
> > Therefore, I keep my original rating. I encourage the authors to re-conduct experiments and re-submit the manuscript to another top-tier conference.

---

> ### Author Response · Authors · 2024-11-28
> **Response to Reviewer F3iL [1/2]**
>
> We appreciate your further constructive comments and suggestions to help refine our paper. We have responded to your concerns as follows.
>
> > **Q1: Small concern in Q2 and Q3. The advantage of this work is not significant.**
>
> **Concern in Q2**. Our method achieves highly competitive results on SPAQ and TID2013, achieving the highest average performance. Further, we evaluate the AIGC image evaluation dataset AIGC2023. Note that the training data and data splitting ratio here are the same as LIQE. We can see that on the AIGC2023 dataset, we achieve a SRCC of 0.770, +2.5 higher than 0.775 of LIQE. This further verifies the strong generalization ability of our model.
>
> | Method          | TID2013 | SPAQ  | AIGC2023 | Average |
> |-----------------|---------|-------|----------|---------|
> | **NIQE**        | 0.314   | 0.578 | -        | 0.446   |
> | **DBCNN$_s$**   | 0.686   | 0.412 | 0.730    | 0.609   |
> | **PaQ2PiQ**     | 0.423   | 0.823 | 0.643    | 0.630   |
> | **MUSIQ$_r$**   | 0.584   | 0.853 | 0.736    | 0.724   |
> | **UNIQUE**      | 0.768   | 0.838 | 0.761    | 0.789   |
> | **LIQE**        | 0.811   | 0.881 | 0.744    | 0.812   |
> | **Gamma$^{+}$** | 0.805 | 0.894 | 0.770    | **0.823** |
>
> **Concern in Q3**. Our model is more efficient and effective than Q-Align. In terms of performance, our method achieves an average SRCC of 0.943 (vs. 0.934) and an average SRCC improvement of 0.949 (vs. 0.938) compared to Q-Align. In terms of efficiency, Q-Align uses a heavy language model with 8.2B (8200M) parameters, while our Gamma has only 122.8M trainable parameters (272.7M in total). Our approach also has lower inference latency (0.1s vs. 0.025s). In practical scenarios, such as using IQA models for AIGC data filtering, our approach is more resource-friendly.
>
> | Dataset | Trainable Parms |Inference time | KonIQ SRCC | KonIQ PLCC | SPAQ SRCC | SPAQ PLCC | KADID SRCC | KADID PLCC | Average SRCC | Average PLCC |
> |---------|------------|------------|-----------|-----------|------------|------------|------------|------------|------------|------------|
> | **Q-Align** | 8.2B (8200M) | 0.1s	 | 0.938      | 0.945      | **0.931** | **0.933** | 0.934      | 0.935      |0.934      | 0.938 |
> | **Gamma$^{+}$** | 122.8M | 0.025s | **0.940**   | **0.950**   | 0.928     | 0.932     | **0.962**   | **0.964**   | **0.943**   | **0.949** |
>
>
> > **Q2: LIQE and UNIQUE all include the BID dataset, which is missing here. Second, the training/test splits in UNIQUE and LIQE are different.**
>
> We add the BID dataset and retrain our model using the same 7:1:2 ratio as LIQE. The results are shown in the table below. Note that we use the results of UNIQUE from LIQE to ensure the same experimental settings. We can observe that our method has the best performance, achieving an average of 0.929 SRCC (vs. 0.922 of LIQE) and 0.941 SRCC (vs. 0.923 of LIQE). We present these results in Section 4.4 and Table 7, marked in blue. The results of the cross-dataset validation are also revised based on the newly trained model.
>
> | Method         | LIVE SRCC | LIVE PLCC | CSIQ SRCC | CSIQ PLCC | KADID SRCC | KADID PLCC | BID SRCC | BID PLCC | LIVEC SRCC | LIVEC PLCC | KonIQ SRCC | KonIQ PLCC | Average SRCC | Average PLCC |
> |----------------|-------------|-------------|-------------|-------------|--------------|--------------|------------|------------|--------------|--------------|--------------|--------------|----------------|----------------|
> | **UNIQUE**     | 0.961       | 0.952       | 0.902       | 0.921       | 0.884        | 0.885        | 0.852      | 0.875      | 0.854        | 0.884        | 0.895        | 0.900        | 0.891          | 0.903          |
> | **LIQE**       | 0.970       | 0.951       | 0.936       | 0.939       | 0.930        | 0.931        | 0.875      | 0.900      | 0.904        | 0.910        | 0.919        | 0.908        | 0.922          | 0.923          |
> | **Gamma$^{+}$** | 0.960    | 0.947       | 0.936       | 0.957       | 0.955        | 0.956        | 0.901      | 0.925      | 0.890        | 0.915        | 0.933        | 0.946        | **0.929**      | **0.941**      |

---

> ### Author Response · Authors · 2024-11-28
> **Response to Reviewer F3iL [2/2]**
>
> > **Q3: Unfair comparison in the main table of the manuscript.**
>
> For LIQE performance, we use the results from the LoDA [1] directly. We also carefully compare the LIQE results in the LoDA paper with the results of the original LIQE paper to ensure that the correct metrics are used. Unfortunately, we do not notice that LIQE uses a different data splitting ratio than us. For comparison in the main table, we retrain LIQE using a data split of 8:2.
>
> [1] Xu K, Liao L, Xiao J, et al. Boosting Image Quality Assessment through Efficient Transformer Adaptation with Local Feature Enhancement[C]//Proceedings of the IEEE/CVF Conference on Computer Vision and Pattern Recognition. 2024: 2662-2672.
>
> $*$ indicates the result of our training. Due to page limitations, we only present part of the Gamma data from the main table.
> | Method         | LIVE SRCC | LIVE PLCC | CSIQ SRCC | CSIQ PLCC | KADID SRCC | KADID PLCC | LIVEC SRCC | LIVEC PLCC | KonIQ SRCC | KonIQ PLCC |
> |----------------|-------------|-------------|-------------|-------------|--------------|--------------|------------|------------|--------------|--------------|
> | **LIQE$*$**       | 0.972       | 0.953       | 0.946       | 0.943       | 0.932        | 0.933        | 0.902      | 0.908      | 0.920        | 0.905        |
> | **Gamma$^{+}$** | 0.953   | 0.953       | 0.960       | 0.968       | 0.962        | 0.964       | 0.891      | 0.914      | 0.939        | 0.950        |
>
> Through the responses to Q2 and Q3, we hope to address your concerns about the comparison experiments in our paper.

---

> ### Author Response · Authors · 2024-12-02
> **To Reviewer F3iL**
>
> Dear Reviewer F3iL,
>
> We truly appreciate your guidance to advance our work. We genuinely value the time and effort you dedicated to reviewing our paper. Regarding your concern about the unfair setting of the experiment, we have conducted detailed experiments. If you have any other questions, we would be more than happy to respond!
>
> Considering that the discussion will end soon, we eagerly look forward to your response.
>
> Best regards,
>
> Authors

---

> > ### Author Response · Authors · 2024-12-03
> > **To Reviewer F3iL**
> >
> > Dear Reviewer F3iL,
> >
> > We truly appreciate your guidance to advance our work. We genuinely value the time and effort you dedicated to reviewing our paper. Considering that the discussion will end soon, we eagerly look forward to your response.
> >
> > We will open source this large-scale pre-trained image evaluation model. It has excellent image evaluation capabilities in a variety of scenarios. Our model has broad applications in various real-world scenarios, such as selecting high-quality images in a data engine, or acting as reward models when aligning image generative models with human feedback. In addition, it can be used as a base model to assist other downstream tasks, such as medical image quality assessment (Table 11b).
> >
> > Best regards,
> >
> > Authors

---

### Official Review · Reviewer_PLKG · 2024-11-04

**Soundness:** 2
**Presentation:** 2
**Contribution:** 2
**Rating:** 5
**Confidence:** 4

**Summary:**

This submission propose a generic image assessment model using mixture of assessment experts, named Gamma. To deal with the problem of applying the image assessment model across various scenarios, Gamma proposes two techniques: 1) proposing a Mixture of Assessment Experts (MoAE) module, which employs shared and adaptive experts to dynamically learn common and specific knowledge for different datasets; 2) introducing a Scene-based Differential Prompt (SDP) strategy, which uses scene-specific prompts to provide prior knowledge and guidance during the learning process. Although the experiments shows the better performance of the proposed method, there are still some concerns for its acceptance.

**Strengths:**

The experiments show the better performance of the proposed method

**Weaknesses:**

1)	The relationship between the adaptive experts and the scene-based differential prompt is unclear. The adaptive experts is also a type of prompt engineering to capture the specific knowledge of the datasets, which is much similar to the scene-based prompt with scene-specific priors. Much analysis on their inner mechanism is suggested to be added.
2)	Furthermore, rather than the statistical results on the datasets, I would like to see the analysis and experiment results to prove that the adaptive experts indeed capture the specific knowledge of different datasets and how the specific knowledge is reflected in the adaptive experts.
3)	Ablation studies include experiments on the number of experts. What is the relationship between their number with the number of the datasets? As shown in Table 2, there are five datasets used for ablation, but three experts get the best performance. Could you help to analyze how the three experts capture the knowledge of the five datasets?

**Questions:**

Please refer to the above comments

---

> ### Author Response · Authors · 2024-11-25
> **Response to Reviewer PLKG [1/2]**
>
> We sincerely appreciate your valuable comments. Your advice significantly helps in enhancing the quality of our work. We have modified the paper based on your valuable comments, marked in blue.
>
> > **Q1: The relationship between the adaptive experts and the scene-based differential prompt is unclear.**
>
> Adaptive Experts dynamically activate the experts to different degrees based on the input image. Scene-based differential prompt uses different prompts for images of different scenes. Both methods can help the model learn different representative representations for different datasets. We conduct ablation experiments on adaptive experts and the scene-based differential prompt to explore their relationship and impact on model performance. The results (table below) show that both methods can improve the performance of the model, such as +7.8% SRCC of SDP and +8.6% SRCC of MoAE on LIVEC. This shows the effectiveness of the adaptive expert feature learning and text guidance for multi-dataset learning. When the two methods are used together, the model will achieve the best results. Therefore, the two methods are mutually beneficial. We supplement these results and analysis in the Section 4.5 (Effectiveness of the prompt strategy) and Table 2.
>
> | MoAE  | SDP | LIVEC SRCC | LIVEC PLCC | KonIQ SRCC | KonIQ PLCC | LIVE SRCC | LIVE PLCC | CSIQ SRCC | CSIQ PLCC | AGIQA3k SRCC | AGIQA3k PLCC | UWIQA SRCC | UWIQA PLCC | AVA SRCC | AVA PLCC |
> |-----------------------|---------|------------|------------|------------|------------|-----------|-----------|-----------|-----------|--------------|--------------|------------|------------|----------|----------|
> | ✗   | ✗|0.765      | 0.792      | 0.858      | 0.885      | 0.927     | 0.918     | 0.852     | 0.898     | 0.800        | 0.866        | 0.750      | 0.768      | 0.681    | 0.672    |
> | ✗   | ✓|0.843      | 0.856      | 0.874      | 0.896      | 0.929     | 0.917     | 0.866     | 0.901     | 0.841        | 0.887        | 0.770      | 0.780      | 0.721    | 0.715    |
> | ✓   | ✗ |0.851      | 0.871      | 0.940      | 0.949      | 0.957     | 0.952     | 0.949     | 0.966     | 0.870        | 0.910        | 0.863      | 0.878      | 0.740    | 0.737    |
> | ✓   | ✓|0.891      | 0.914      | 0.939      | 0.950      | 0.960     | 0.968     | 0.953     | 0.953     | 0.887        | 0.923        | 0.873      | 0.884      | 0.750    | 0.749    |
>
> ---
>
> > **Q2: The analysis and experiment results to prove that the adaptive experts indeed capture the specific knowledge of different datasets and how the specific knowledge is reflected in the adaptive experts.**
>
> Different experts learn domain-specific features of different datasets, which effectively addresses their content differences and label biases. To explore the preferences of different experts for different datasets, in addition to the statistical analysis on the dataset, we add an experiment in which we only use one adaptive expert and set the router weights of the other experts to 0. As shown in Table below, the first expert performs well on most datasets, indicating it learns a general image assessment ability. The second and third experts focus on AIGC IQA and IAA tasks, respectively, and the third expert also shows excellent evaluation capabilities for natural images. These results indicate that different experts have learned domain-specific features of different datasets. They collaborate to achieve the powerful image assessment model Gamma. We add these analyses and results to Section 4.5 (Analysis of the adaptive experts) and Table 10.
>
> | Dataset       | LIVEC SRCC | LIVEC PLCC | KonIQ SRCC | KonIQ PLCC | LIVE SRCC | LIVE PLCC | CSIQ SRCC | CSIQ PLCC | AGIQA3k SRCC | AGIQA3k PLCC | UWIQA SRCC | UWIQA PLCC | GFIQA SRCC | GFIQA PLCC | AVA SRCC | AVA PLCC |
> |---------------|------------|------------|------------|------------|-----------|-----------|-----------|-----------|--------------|--------------|------------|------------|------------|------------|----------|----------|
> | **1-th expert** | **0.847**  | **0.860**  | **0.927**  | **0.938**  | **0.933** | **0.933** | **0.894** | **0.906** | 0.815        | 0.870        | **0.770**  | **0.779**  | **0.959**  | **0.957**  | 0.666    | 0.673    |
> | **2-th expert** | 0.715      | 0.672      | 0.681      | 0.717      | 0.900     | 0.861     | 0.815     | 0.846     | **0.832**    | **0.885**    | 0.755      | 0.756      | 0.826      | 0.797      | 0.663    | 0.652    |
> | **3-th expert** | 0.768      | 0.741      | 0.794      | 0.818      | 0.918     | 0.917     | 0.833     | 0.877     | 0.808        | 0.910        | 0.691      | 0.709      | 0.903      | 0.897      | **0.715**| **0.716**|
> | **Gamma**      | 0.851      | 0.871      | 0.940      | 0.949      | 0.957     | 0.952     | 0.949     | 0.966     | 0.870        | 0.910        | 0.863      | 0.878      | 0.970      | 0.970      | 0.740    | 0.737    |

---

> ### Author Response · Authors · 2024-11-25
> **Response to Reviewer PLKG [2/2]**
>
> > **Q3: What is the relationship between the number of experts and the number of datasets? As shown in Table 2, there are five datasets used for ablation. How the three experts capture the knowledge of the five datasets?**
>
> - The number of experts is related to the overall distribution of the data. We found that using 3 experts achieves excellent results and adding more experts does not significantly improve performance through ablation experiments. In addition, in other areas such as large language models [1-2], the number of experts in the MoE structure is also usually set empirically.
> - Note that we do not use 5 datasets for ablation, but uniformly 12 datasets. The performance of different models on some datasets is saturated, resulting in similar results. Therefore, considering the page limit, we only show the datasets with relatively large differences in results.
> - We do not set one expert for each dataset. In fact, we use a soft router, i.e., each expert is assigned a weight, and the sum of all expert weights is 1. For different datasets, the model activates these experts to different degrees adaptively based on the input image. Therefore, different weight combinations make our expert activation patterns diverse, thus our MoAE with three adaptive experts can handle many datasets.
>
> [1] Dai D, Deng C, Zhao C, et al. Deepseekmoe: Towards ultimate expert specialization in mixture-of-experts language models[J]. arXiv preprint arXiv:2401.06066, 2024.
>
> [2] Jiang A Q, Sablayrolles A, Roux A, et al. Mixtral of experts[J]. arXiv preprint arXiv:2401.04088, 2024.

---

> ### Author Response · Authors · 2024-12-02
> **To Reviewer PLKG**
>
> Dear Reviewer PLKG,
>
> We truly appreciate your guidance to advance our work. We genuinely value the time and effort you dedicated to reviewing our paper. Considering that the discussion will end soon, we eagerly look forward to your response.
>
> Best regards,
>
> Authors

---

> ### Author Response · Authors · 2024-12-03
> **To Reviewer PLKG**
>
> Dear Reviewer PLKG,
>
> We truly appreciate your guidance to advance our work. We genuinely value the time and effort you dedicated to reviewing our paper. Considering that the discussion will end soon, we eagerly look forward to your response.
>
> We will open source this large-scale pre-trained image evaluation model. It has excellent image evaluation capabilities in a variety of scenarios. Our model has broad applications in various real-world scenarios, such as selecting high-quality images in a data engine, or acting as reward models when aligning image generative models with human feedback. In addition, it can be used as a base model to assist other downstream tasks, such as medical image quality assessment (Table 11b).
>
> Best regards,
>
> Authors

---

### Note · Authors · 2025-01-23

I have read and agree with the venue's withdrawal policy on behalf of myself and my co-authors.